# NoisyGRPO: Incentivizing Multimodal CoT Reasoning via Noise Injection and Bayesian Estimation

**Longtian Qiu**[1], **Shan Ning**[1,3], **Jiaxuan Sun**[1], **Xuming He**[1,2]

[1]ShanghaiTech University, Shanghai, China
[2]Shanghai Engineering Research Center of Intelligent Vision and Imaging
[3]Lingang Laboratory, Shanghai, China
{qiult, ningshan2022, sunjx2022, hexm}@shanghaitech.edu.cn

## Abstract

Reinforcement learning (RL) has shown promise in enhancing the general Chain-of-Thought (CoT) reasoning capabilities of multimodal large language models (MLLMs). However, when applied to improve general CoT reasoning, existing RL frameworks often struggle to generalize beyond the training distribution. To address this, we propose NoisyGRPO, a systematic multimodal RL framework that introduces controllable noise into visual inputs for enhanced exploration and explicitly models the advantage estimation process via a Bayesian framework. Specifically, NoisyGRPO improves RL training by: (1) **Noise-Injected Exploration Policy**: Perturbing visual inputs with Gaussian noise to encourage exploration across a wider range of visual scenarios; and (2) **Bayesian Advantage Estimation**: Formulating advantage estimation as a principled Bayesian inference problem, where the injected noise level serves as a prior and the observed trajectory reward as the likelihood. This Bayesian modeling fuses both sources of information to compute a robust posterior estimate of trajectory advantage, effectively guiding MLLMs to prefer visually grounded trajectories over noisy ones. Experiments on standard CoT quality, general capability, and hallucination benchmarks demonstrate that NoisyGRPO substantially improves generalization and robustness, especially in RL settings with small-scale MLLMs such as Qwen2.5-VL 3B. The project page is available at `https://artanic30.github.io/project_pages/NoisyGRPO`.

## 1 Introduction

The success of OpenAI's O1 [34] and DeepSeek-R1 [7] highlights that activating Chain-of-Thought (CoT) reasoning abilities through Reinforcement Learning (RL) is a promising strategy for scaling model intelligence at test time. In particular, value-free RL methods such as Group Relative Policy Optimization [44] (GRPO) have demonstrated substantial improvements in LLM performance while avoiding the costly requirement of training a separate value model. As recent works show [38, 13, 8, 30, 46], this efficiency in both data and computation makes GRPO an appealing approach for extension to complex visual language tasks such as visual math problem solving.

However, when applying GRPO to improve general chain-of-thought (CoT) reasoning in multimodal large language models (MLLMs), we observe notable limitations in the model's ability to generalize beyond the training distribution, which hinders the broad applicability of MLLMs. As shown in Figure 1, higher training rewards under the GRPO framework do not consistently yield better evaluation performance. We identify two primary factors contributing to this issue: (1) *Limited Policy Exploration.* RL in the generative model domain utilizes temperature sampling to generate multiple rollouts for policy exploration; however, as noted in DAPO [58], these rollouts often converge to near-identical outputs during training, leading to insufficient exploration and early deterministic

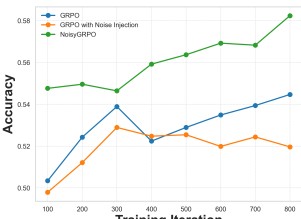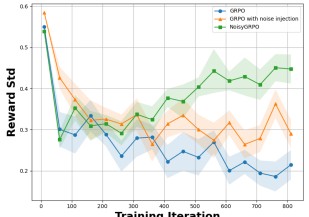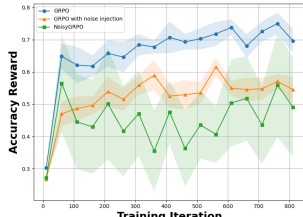

Figure 1: **Performance and training statistics of three RL methods.** *GRPO with noise injection* refers to GRPO trained with noise-perturbed rollouts. The left plot shows evaluation performance on the MMStar benchmark over training iterations. The middle plot presents the standard deviation of rewards, reflecting the exploration degree of the policy. The right plot shows the accuracy reward, indicating how well the model fits the training data. Shaded areas represent the variance of the corresponding metrics.

policy. (2) *Missing process supervision.* Rule-based rewards judge only the final answer, leaving the intermediate reasoning steps unsupervised; the policy, therefore, learns shortcuts and visual hallucinations that do not transfer out of distribution.

To address these challenges, we propose **NoisyGRPO**, a unified framework that enhances multimodal reinforcement learning, such as GRPO [44] by both promoting thorough exploration and ensuring robust policy optimization. Our key innovation is to inject controllable noise into visual inputs during rollouts, thereby encouraging diverse exploration and providing an unbiased, trajectory-level measure of reasoning difficulty. However, while promoting exploration, noise injection induces an off-policy scenario: the behavior policy that collects trajectories generated with noise-disturbed images differs from the target policy that operates on clean visual input. This distribution shift causes the value function to be queried on out-of-distribution states, which introduces bias into the advantage estimates and corrupts the policy gradient, as illustrated in Figure 1. To tackle this issue, we introduce a Bayesian advantage estimation framework that treats the injected noise as a prior and the trajectory reward as the likelihood within a principled Bayesian formulation, which allows us to explicitly calibrate the policy update according to trajectory-level noise and reward signals.

NoisyGRPO consists of two key components without additional computational overhead. First, **Noise Injection** augments policy exploration by applying Gaussian noise of various magnitudes (sampled from a uniform distribution) to input images in rollouts, maintaining trajectory diversity (indicated by the consistently high *Reward Std* shown in Figure 1) and reducing hallucinations via implicitly favoring visually grounded reasoning. The second component is **Bayesian Advantage Estimation**, which integrates the magnitude of the injected noise with observed trajectory rewards to yield more accurate advantage estimates. Specifically, we treat the likelihood uncertainty as a fixed constant while modeling the prior uncertainty as a function of the variance among rewards within the same trajectory group. The intuition is that if trajectory rewards are similar despite varying noise magnitudes, the prior uncertainty remains high; in contrast, larger reward variation across noise levels corresponds to reduced prior uncertainty. By defining advantage estimation in this Bayesian manner, we obtain a posterior that adaptively incorporates both noise magnitude and trajectory reward statistics, leading to more accurate and robust advantage estimates.

To demonstrate NoisyGRPO's effectiveness, we evaluate its impact on improving MLLMs' *general capabilities*, which is a more challenging setting than addressing domain-specific problems. Specifically, we evaluate NoisyGRPO from three dimensions: CoT quality [16], comprehensive MLLM capability [5, 63], and hallucination [50]. Results show that NoisyGRPO consistently outperforms GRPO, particularly with small-scale MLLM—for instance, achieving a *+4.4* improvement in CoT quality on MME-COT and a *+3.7* gain in average performance on MMStar with Qwen2.5-VL 3B.

Our main contributions are summarized as follows:

- We propose NoisyGRPO, a reinforcement learning framework designed to enhance the multimodal Chain-of-Thought (CoT) reasoning capabilities of MLLMs on general tasks.

- NoisyGRPO employs a noise injection strategy to promote policy exploration, and introduces a Bayesian advantage estimation method that mitigates the negative effects of noise by combining prior estimates from noise magnitude with observed response correctness.

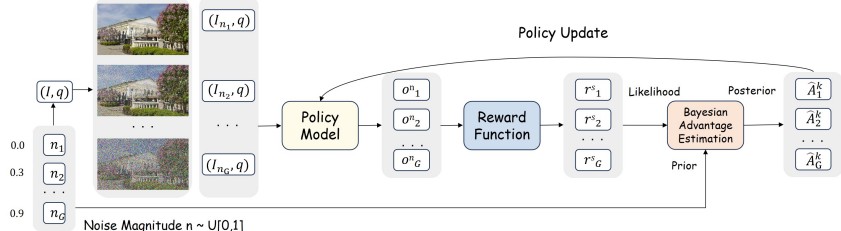

Figure 2: **Overall pipeline of NoisyGRPO.** For each image-question pair, we sample noise and inject it into the image. The policy model generates rollouts based on the perturbed inputs, and the reward function evaluates them. We then compute the posterior advantage by combining the noise-based prior with the reward-based observation.

- Experimental results show that NoisyGRPO consistently outperforms GRPO across CoT quality, general capability, and hallucination benchmarks, with especially strong gains in RL with small-scale MLLMs.

## 2   Related Works

**Multi-modal Large Language Model**   Vision-language models (VLMs) have rapidly advanced in their ability to comprehend and reason over both visual and textual modalities and demonstrate great improvements over downstream tasks [35, 64, 12, 9, 31, 40]. These models typically integrate visual encoders with large language models (LLMs) to enable cross-modal understanding and inference. Foundational works such as Flamingo [2] and BLIP-2 [20] established effective strategies for aligning vision and language components. The LLaVA-series [25, 24, 18] and SPHINX-series [22, 10] further advanced the field by introducing visual instruction tuning, significantly enhancing multimodal capabilities. Large-scale models like GPT-4o [14] and Gemini [42] have demonstrated strong general-purpose visual understanding through large-scale multimodal pertaining. To address scalability, mixture-of-experts approaches—such as DeepSeek-VL2 [55], Uni-MoE [21], and MoVA [65]—improve computational efficiency by selectively activating expert modules based on input characteristics. More recently, domain-specific methods such as Math-LLaVA [47] and MAVIS [60] employed mathematical visual instruction tuning to improve VLMs' ability to interpret and solve complex multimodal math problems. At the same time, unified architectures like SEED-X [11], Chameleon [48], Show-O [56], and the Janus series [54, 29, 6] integrate both visual understanding and generation capabilities within a single framework. Despite these advances, most existing VLMs still struggle with robust visual reasoning, particularly in tasks that demand deep visual analysis and complex multi-step reasoning.

**Reinforcement Learning in LLMs and VLMs**   Reinforcement Learning (RL) has emerged as a key technique for enhancing the capabilities of Large Language Models (LLMs) and Multimodal Large Language Models (MLLMs). Early advancements focused on Reinforcement Learning from Human Feedback (RLHF) [37], which aligns model outputs with human preferences [1]. More recent research has demonstrated that RL-based methods can significantly improve reasoning abilities. For instance, DeepSeek-R1 [7] integrates rule-based rewards with Group Relative Policy Optimization (GRPO) [45], while Kimi-1.5 [49] employs a variant of online policy mirror descent—both achieving notable gains in reasoning performance. In the multimodal domain, RL has also shown promise. Direct Preference Optimization (DPO) [41], a simple yet effective RL method, has been applied to reduce hallucination in MLLMs [39, 36, 61]. GRPO has also been extended to vision-language models for specialized tasks such as geometry reasoning and object counting [38, 13, 8, 30, 46]. Though these works apply the GRPO framework to multimodal tasks, their focuses differ from ours. For example, VLM-R1 [46] adapts it to detection tasks with task-specific rewards. In contrast, we propose a principled framework that systematically improves multimodal RL algorithms like GRPO in multimodal settings by enhancing both policy exploration and advantage estimation.

## 3   NoisyGRPO

In this section, we introduce NoisyGRPO, a reinforcement learning framework designed to enhance the Chain-of-Thought reasoning capabilities of multimodal large language models (MLLMs). To

address the generalization limitations of vanilla GRPO when applied to MLLMs, NoisyGRPO encourages policy exploration via perturbing visual inputs and mitigates the resulting adverse effects through Bayesian advantage estimation. Specifically, in section 3.1, we present the training objective of NoisyGRPO and highlight its improvements over the standard GRPO. In section 3.2, we introduce the exploration policy with injection noise on the visual input strategy. In section 3.3, we describe the Bayesian advantage estimation method, which incorporates both the level of noise and the trajectory reward to yield more accurate policy gradients. We provide an overall pipeline in Figure 2.

## 3.1 Training Objective and Reward Function

**Training Objective**   To highlight the key innovations of NoisyGRPO, we start by presenting its training objective and comparing it with vanilla GRPO [44]. NoisyGRPO differs in two main aspects: (1) *Exploration Policy*. We introduce visual noise during sampling to encourage more diverse trajectory exploration. (2) *Advantage Estimation*. By modeling advantage estimation as a Bayesian inference problem, we combine the noise level and observed trajectory reward to compute a more accurate advantage for each trajectory.

Specifically, given a sample $(q, a, I)$ drawn from the dataset $\mathcal{D}$—where $q$ is the input query, $a$ is the ground-truth answer, and $I$ is the corresponding image—we sample $G$ rollouts $\{o_i^n\}_{i=1}^{G}$ using a noise-injected exploration policy $\pi^{\text{noise}}$, where each $o_i^n$ is sampled under a different noise level applied to the image. The corresponding group-wise advantage $\tilde{A}_i^k$ is estimated based on both trajectory reward and noise level. The training objective of NoisyGRPO is defined as follows:

$$J_{\text{NoisyGRPO}}(\theta) = \mathbb{E}_{(q,a,I)\sim\mathcal{D},\ \{o_i^n\}_{i=1}^{G}\sim\boldsymbol{\pi}_{\theta_{\text{old}}}^{\text{noise}}(\cdot|q,I)}$$

$$\left[ \frac{1}{G} \sum_{i=1}^{G} \min\left( \frac{\pi_\theta(o_i^n \mid q, I)}{\pi_{\theta_{\text{old}}}(o_i^n \mid q, I)} \tilde{\boldsymbol{A}}_{\boldsymbol{i}}^{\boldsymbol{k}},\ \text{clip}\left( \frac{\pi_\theta(o_i^n \mid q, I)}{\pi_{\theta_{\text{old}}}(o_i^n \mid q, I)}, 1 - \epsilon, 1 + \epsilon \right) \tilde{\boldsymbol{A}}_{\boldsymbol{i}}^{\boldsymbol{k}} \right) - \beta D_{\text{KL}}(\pi_\theta \| \pi_{\text{ref}}) \right],$$
(1)

where $\pi_\theta$ and $\pi_{\theta_{\text{old}}}$ are the current and previous policies respectively, $\epsilon$ is the PPO clipping parameter, $\beta$ is the KL penalty coefficient, $\pi_{\text{ref}}$ is a reference policy, and $D_{\text{KL}}$ denotes KL divergence. The modifications from vanilla GRPO are marked in **bold**—specifically, the exploration policy $\boldsymbol{\pi}^{\text{noise}}$ and the group-relative advantage $\tilde{\boldsymbol{A}}_{\boldsymbol{i}}^{\boldsymbol{k}}$. We elaborate on these components in the following sections.

**Reward Function**   For the reward function, we adopt two types of rule-based rewards inspired by GRPO: *Accuracy* and *Format rewards*. The Correctness reward measures whether the model's final answer is accurate, regardless of the correctness of intermediate CoT steps. The Format reward encourages the model to produce answers in the expected CoT style. It is worth noting that our training data contains many open-ended VQA samples to improve general CoT ability in multimodal settings. Unlike code or math tasks with short, matchable answers, these samples require a more flexible reward. We use a text embedding model such as SBERT [43] to assess semantic similarity for correctness. Although this may introduce embedding bias, our advantage estimation 3.3 helps mitigate it. In conclusion, the reward is defined as follows:

- **Accuracy rewards**: We utilize the SBERT model to compute embeddings for both the predicted and ground-truth answers, and define their similarity as the accuracy reward. To filter low-confidence cases, we set the reward to 0 when the similarity is below $\tau$. For yes/no and multiple-choice questions, we apply exact match: a correct match receives a reward of 1, otherwise 0.

- **Format rewards**: To ensure CoT reasoning, we introduce a format reward that requires the model's reasoning process to appear between <think> and </think> tags.

## 3.2 Noise Injected Exploration Policy

To enhance the generalization of the multimodal RL framework, we introduce a noise-injected exploration strategy to construct our exploration policy $\pi^{\text{noise}}$, as shown in Figure 2. Different from injecting noise into the entire policy model as in [15], we adopt a targeted and structured strategy by applying multi-step Gaussian noise to the input image $I$ based on a diffusion-style forward process. For each rollout, we first sample a noise level $n_i \sim \mathcal{U}(0, 1)$, representing a normalized noise step. We then perturb $I$ using the corresponding diffusion process at timestep $\lfloor n_i \cdot T \rfloor$, where $T$ is the total number of diffusion steps. $n_i = 0$ yields the original clean image, while $n_i = 1$ corresponds

to the image at the final diffusion step, effectively resembling pure Gaussian noise. This produces a noisy image $I_{n_i}$ which is then used to generate the rollout $o_i^n$ under the exploration policy $\pi^{\text{noise}}$, resulting in a diverse set of responses $\{o_i^n\}_{i=1}^G$ conditioned on different noise intensities. Notably, noise injection is applied only during rollout collection to encourage exploration. Clean inputs are used during policy updates to ensure stable learning and preserve the accuracy of the learned policy $\pi_\theta$.

### 3.3 Bayesian Advantage Estimation

**Bayesian Modeling of Trajectory Quality**  After collecting rollouts using the noise-injected exploration policy, we formulate the trajectory quality as a latent variable $r_i$, which represents the underlying quality of a sampled rollout $o_i$ obtained under noise-injected exploration. To integrate both prior knowledge from noise level and observation, such as the trajectory reward, we adopt a Bayesian modeling approach, treating $r_i$ as a Gaussian random variable. Specifically, we assume:

$$r_i \sim \mathcal{N}(\mu_i, \sigma^2), \tag{2}$$

where $\mu_i$ denotes the mean of the trajectory quality $r_i$ and $\sigma^2$ quantifies the uncertainty. To update this prior based on observed evidence, we consider a Bayesian inference framework. Let $\tilde{r}_i$ be an observed variable that serves as a noisy indicator of the true quality $r_i$. We assume the observation model is Gaussian:

$$\tilde{r}_i \sim \mathcal{N}(r_i, \sigma_s^2), \tag{3}$$

where $\sigma_s^2$ captures the reliability of the observation process. Under this setup, the posterior distribution of $r_i$ given $\tilde{r}_i$ remains Gaussian, and the posterior mean can be derived analytically as:

$$\hat{r}_i = \tilde{r}_i + \frac{\sigma_s^2}{\sigma^2 + \sigma_s^2}(\mu_i - \tilde{r}_i). \tag{4}$$

This posterior mean balances the prior distribution and the observation, weighted inversely by their respective variances. Intuitively, more confident observations (small $\sigma_s^2$) shift the posterior closer to $\tilde{r}_i$, while a more confident prior (small $\sigma^2$) keeps the posterior closer to $\mu_i$.

**Advantage Estimation**  For a set of samples $\{(n_i, o_i, r_i^s)\}_{i=1}^G$, where $n_i$ is the injected noise level, $o_i$ is the generated trajectory, and $r_i^s$ is the trajectory semantic reward, which combines both accuracy and format reward. For the prior estimation of trajectory quality $r_i$, we set the noisy reward as $r_i^n = 1 - n_i$ and treat it as the prior estimation. The observation is the semantic reward $r_i^s$. We apply normalization to make the prior and semantic rewards comparable across the group of $G$ samples. Given a set of scalar values $\{x_j\}_{j=1}^G$, the normalization operation is defined as:

$$\text{Norm}(x_i, \{x_j\}_{j=1}^G) = \frac{x_i - \text{mean}(\{x_j\}_{j=1}^G)}{\text{std}(\{x_j\}_{j=1}^G)}, \tag{5}$$

We apply this normalization to both the prior and semantic rewards as follows:

$$\hat{r}_i^n = \text{Norm}(r_i^n, \{r_j^n\}_{j=1}^G), \quad \hat{r}_i^s = \text{Norm}(r_i^s, \{r_j^s\}_{j=1}^G), \tag{6}$$

For clarity, we assume both normalized signals are corrupted by a Gaussian, which serves as a core assumption in our Bayesian estimation framework. An analysis of this assumption and its empirical justification is provided in the Appendix. Accordingly, we model the latent reward $r_i$ as a random variable drawn from a Gaussian prior centered at $\hat{r}_i^n$, while the observed semantic reward $\hat{r}_i^s$ is treated as a noisy observation of $r_i$:

$$r_i \sim \mathcal{N}(\hat{r}_i^n, \sigma_n^2), \qquad \hat{r}_i^s \sim \mathcal{N}(r_i, \sigma_s^2). \tag{7}$$

To this end, the key challenge is to determine the uncertainty of both signals, which is represented by the $\sigma_n^2$ and $\sigma_s^2$. We fix $\sigma_s^2$ to a constant $\alpha$, as the semantic reward is computed using a text embedding model whose bias is relatively stable. To modulate the prior uncertainty, we define the prior variance $\sigma_n^2$ to be inversely related to the variability of semantic rewards within the group. The underlying intuition is that $\hat{r}_i^s$ and $\hat{r}_i^n$ are positively correlated, as both reflect the underlying trajectory quality, and $n_i$ is sampled from a uniform distribution. Therefore, when the group-wise variance of $\hat{r}_i^s$ is small, the certainty of the prior distribution is low. Specifically:

$$\sigma_s^2 = \alpha, \qquad \sigma_n^2 = \frac{\gamma}{\gamma + (\text{std}(\{r_i^s\}_{i=1}^G))^2}, \tag{8}$$

Table 1: **Results on CoT quality benchmark MME-CoT.** We report results across three evaluation dimensions of CoT quality to comprehensively characterize the performance of NoisyGRPO.

| | CoT quality | | | CoT Robustness | | | CoT Efficiency | | |
|---|---|---|---|---|---|---|---|---|---|
| | F1 | Precision | Recall | Avg. | Stability | Efficacy | Avg. | Rel. Rate | Ref. Quality |
| LLaVA-OV-7B | 30.9 | 50.9 | 22.2 | -3.4 | -3.8 | -3.0 | 91.5 | 83.0 | 100.0 |
| LLaVA-OV-72B | 36.3 | 57.3 | 26.6 | -0.2 | 0.3 | -0.6 | 95.4 | 90.8 | 100.0 |
| MiniCPM-V-2.6 | 39.8 | 57.3 | 30.5 | -3.5 | -4.8 | -2.2 | 92.8 | 85.7 | 100.0 |
| InternVL2.5-8B-MPO | 43.0 | 60.4 | 33.4 | 0.6 | 0.3 | 0.9 | 94.7 | 89.3 | 100.0 |
| Qwen2-VL-72B | 56.2 | 77.3 | 44.2 | -2.1 | -6.5 | 2.4 | 96.5 | 92.9 | 100.0 |
| GPT-4o | 64.0 | 85.4 | 51.2 | 2.1 | -1.0 | 5.1 | 96.0 | 92.0 | 100.0 |
| Kimi k1.5 | 64.2 | 92.0 | 49.3 | 1.4 | 2.9 | 0.0 | 82.2 | 92.2 | 72.2 |
| *Reasoning Models with Qwen2.5-VL 3B* | | | | | | | | | |
| GRPO | 36.2 | 53.0 | 27.4 | -7.0 | -7.5 | -6.5 | 51.1 | **76.3** | 26.0 |
| NoisyGRPO | **40.6** | **57.3** | **31.5** | **-2.4** | **-2.4** | **-2.4** | **55.4** | 71.9 | **38.8** |
| *Reasoning Models with Qwen2.5-VL 7B* | | | | | | | | | |
| GRPO | 46.8 | 64.2 | 36.9 | 1.6 | 3.8 | -0.6 | **66.4** | **87.2** | **45.7** |
| NoisyGRPO | **47.7** | **64.8** | **37.7** | **3.9** | **7.8** | **-0.1** | 59.7 | 81.8 | 37.5 |

where $\gamma$ is a scale hyperparameter that modulates the influence of semantic reward variability on prior uncertainty. This formulation implies that when the semantic reward varies widely (i.e., larger standard deviation), the prior distribution is considered more reliable and thus assigned a smaller variance. Given the prior and observation, we apply a Bayesian update to obtain the posterior mean reward:

$$\hat{r}_i = \hat{r}_i^s + \frac{\sigma_s^2}{\sigma_n^2 + \sigma_s^2}(\hat{r}_i^n - \hat{r}_i^s). \tag{9}$$

Finally, to compute the trajectory-wise relative advantage within each group, we normalize the posterior reward:

$$\tilde{A}_i^k = \text{Norm}(\hat{r}_i, \{\hat{r}_j\}_{j=1}^G). \tag{10}$$

## 4 Experiments

### 4.1 Training Data

To enhance the general Chain-of-Thought (CoT) reasoning ability of MLLMs through reinforcement learning, we adopt the visual question answering (VQA) portion of the MM-RLHF [62] training set as our training data. This dataset spans diverse domains, including conversations, safety, multiple-choice questions, captions, and commonsense reasoning. MM-RLHF applies clustering and filtering techniques to curate a high-quality visual instruction-following dataset. In total, it contains 13k VQA samples: 1.2k yes/no questions, 1.3k multiple-choice questions, and 10k open-ended VQA questions. Notably, most of the training data lack structured answers; therefore, we use a text embedding model to compute the accuracy reward. While this may introduce some model bias, our Bayesian advantage estimation helps mitigate its impact.

### 4.2 Implementation Details

We implement the NoisyGRPO based on the VLM-R1 [46] reinforcement learning training framework. For the policy model, we choose the state-of-the-art MLLMs Qwen2.5-VL [4] 3B and 7B. For hyperparameters, we follow the default settings provided by VLM-R1 except for the following changes. The number of sampled rollouts $G$ is set to 4 due to computational resource considerations. In noise injection, the upper bound $\sigma$ for the $\mathcal{U}(0, \sigma)$ is set to 1. The $\alpha$ and $\gamma$ are set to 0.1 and 0.01. The threshold $\tau$ for the accuracy reward is set to 0.6. As there is no validation set, we choose all the hyperparameters based on the results on MMStar [5]. The training takes 6 hours for the 3B variant and 7 hours for the 7B on a single node with 8A100 GPUs.

### 4.3 Evaluation Benchmarks

To comprehensively evaluate the CoT reasoning capabilities of NoisyGRPO, we select benchmarks from three perspectives: (1) **CoT Quality Evaluation**. We use the MME-CoT [16] benchmark, which assesses the quality, robustness, and efficiency of multi-modal CoT reasoning. The benchmark

Table 2: **Performance Comparison on General, Hallucination, and Real-World Benchmarks.**
The *CP, FP, IR, LR, ST, MA, and AVG* denote *Coarse Perception, Fine-grained Perception, Instance Reasoning, Logical Reasoning, Science & Technology, Math, and the average performance*, respectively. The symbols ↑ and ↓ indicate that higher or lower values are preferred. *AMB. Gen.* and *AMB. Dis.* represent the generative and discriminative components of the AMBER benchmark.

| | MMStar | | | | | | | AMB.$^G$ | | | | AMB.$^D$ | MMERW |
|---|---|---|---|---|---|---|---|---|---|---|---|---|---|
| | CP. | FP. | IR. | LR. | ST. | MA. | Avg. | $C_s$ ↓ | Cov. ↑ | Hal. ↓ | Cog. ↓ | F1 ↑ | Acc |
| LLaVA-1.5 7B | 58.8 | 24.0 | 38.8 | 24.0 | 13.6 | 22.8 | 30.3 | 7.8 | 51.0 | 36.4 | 4.2 | 74.7 | - |
| Qwen-VL-Chat | 59.6 | 32.0 | 50.8 | 29.2 | 22.0 | 31.6 | 37.5 | 5.5 | 49.4 | 23.6 | 1.9 | - | - |
| CogVLM-Chat | 66.8 | 36.8 | 49.2 | 31.2 | 23.6 | 11.6 | 36.5 | 5.6 | 57.2 | 23.6 | 1.3 | - | - |
| InternLM-XComposer2 | 70.8 | 48.8 | 65.2 | 56.4 | 42.0 | 49.2 | 55.4 | - | - | - | - | - | - |
| LLaVA-Next 34B | 66.4 | 52.0 | 62.4 | 46.0 | 32.4 | 53.6 | 52.1 | - | - | - | - | - | - |
| GPT4V | 76.6 | 51.4 | 66.6 | 55.8 | 42.6 | 49.8 | 57.1 | 4.6 | 67.1 | 30.7 | 2.6 | - | - |
| Main Results | | | | | | | | | | | | | |
| Qwen2.5-VL 3B | **70.3** | 49.1 | 59.6 | 55.1 | 38.5 | 59.7 | 55.4 | 7.6 | 69.9 | 55.7 | 5.8 | 89.6 | 42.1 |
| + SFT | **70.3** | 49.6 | 59.0 | 55.4 | 39.2 | 60.4 | 55.7 | 6.9 | **70.4** | 48.8 | 3.9 | 88.8 | **44.0** |
| + GRPO | 68.0 | 43.3 | 60.4 | 57.0 | **40.0** | 58.0 | 54.5 | 6.7 | 68.5 | 44.6 | 4.2 | 89.2 | 40.8 |
| + NoisyGRPO | 69.5 | **53.2** | **66.6** | **60.7** | 37.1 | **62.3** | **58.2** | **6.6** | 67.7 | **44.3** | **3.4** | **90.3** | **44.0** |
| Qwen2.5-VL 7B | 73.3 | 60.1 | **72.2** | 62.0 | 44.6 | 67.3 | 63.2 | 4.8 | 63.6 | 27.5 | 1.6 | 87.4 | 43.6 |
| + SFT | 71.8 | 58.8 | 70.2 | 60.8 | 45.2 | 67.4 | 62.4 | 5.4 | **64.9** | 32.4 | 2.1 | 87.5 | 43.5 |
| + GRPO | **73.7** | 57.2 | 70.5 | 64.2 | 44.1 | 65.7 | 62.6 | 4.9 | 64.3 | 28.9 | 1.7 | 88.0 | 42.3 |
| + NoisyGRPO | 72.8 | **63.7** | 71.7 | **66.9** | **48.3** | **71.4** | **65.8** | **4.2** | 63.2 | **23.9** | **1.2** | **88.2** | **44.6** |

covers six categories of questions—including math, OCR, logic, science, space-time, and general scene—providing a holistic view of CoT performance. (2) **General Capability Evaluation**. We adopt the MMStar [5] benchmark, which is constructed by carefully selecting high-quality samples from existing datasets such as AI2D [17], SEED-Bench [19], ScienceQA [28], MMBench [26], MathVista [27], and MMMU [59]. By removing low-quality samples, MMStar provides a cleaner and more comprehensive testbed across six reasoning domains. (3) **Hallucination Evaluation**. We employ the AMBER [50] benchmark to evaluate both the generative and discriminative hallucination behaviors of the model, offering a thorough analysis of output reliability. Additionally, we include MME-RealWorld-Lite [63] to assess VQA performance in real-world scenarios.

## 4.4 Baselines

To fairly demonstrate the effectiveness of NoisyGRPO, we compare it against strong baselines across all evaluation settings. For the CoT quality benchmark, we compare NoisyGRPO with the GRPO [44] algorithm under different scales of Qwen2.5-VL [4], and also report results from a wide range of MLLMs [18, 57, 6, 51, 33, 49, 23, 3, 52, 53, 24, 32, 4], from 7B to 72B models, as well as proprietary models like GPT-4o [33] and Kimi K1.5 [49], for reference. For the general reasoning and hallucination benchmarks, in addition to comparisons between NoisyGRPO and GRPO, we include SFT (Supervised Fine-Tuning) baselines to assess the quality and impact of training data. We also report the performance of state-of-the-art MLLMs for broader comparison.

## 4.5 Performance Analysis

**Benchmark Results**   For the CoT quality evaluation, we report the results of NoisyGRPO and baselines on the MME-CoT [16] benchmark in Table 1. We observe that NoisyGRPO consistently outperforms GRPO on most CoT quality metrics, with particularly significant improvements for the 3B model. Notably, NoisyGRPO also enhances CoT robustness—for example, NoisyGRPO-7B achieves higher robustness scores than proprietary models like GPT-4o [33] and Kimi K1.5 [49]. However, we observe that vanilla GRPO surpasses NoisyGRPO in CoT efficiency when using Qwen2.5-VL 7B. We attribute this to NoisyGRPO's design, which prioritizes grounded reasoning over reasoning efficiency.

For the general reasoning and hallucination benchmarks, Table 2 presents a comparison between NoisyGRPO and baseline methods. The results demonstrate that NoisyGRPO effectively enhances MLLM performance across multiple domains by strengthening CoT reasoning, achieving notable gains such as +2.8 and +2.6 in average MMStar [5] scores over the base model. However, we observe a performance drop in the *Coarse Perception* category for both model variants. We attribute this to the injected noise being insufficient to meaningfully perturb the visual cues necessary for coarse-grained

perception tasks, leading to suboptimal performance for this specific task. Furthermore, we observe that NoisyGRPO performs worse on the *Cov.* metric of the AMBER [50] benchmark, suggesting that the generated captions do not fully cover all elements of the image. We attribute this to NoisyGRPO's emphasis on grounded responses, which tends to produce more concise answers and, consequently, reduces overall coverage.

We also find that supervised fine-tuning (SFT) on the current 13k dataset fails to bring consistent performance gains, highlighting that the improvements stem from reinforcement learning rather than high-quality data alone. Furthermore, the performance improvements on MME-RealWorld-Lite [63] further demonstrate the potential of NoisyGRPO for real-world applications.

**Performance Over Training Iterations**  In addition to reporting the best model performance across various benchmarks, we further analyze the training dynamics of NoisyGRPO to provide a more comprehensive understanding of its behavior. Specifically, we examine the relationship between training iterations and performance on the MMStar benchmark, which evaluates MLLMs across six distinct reasoning dimensions. As shown in Figure 3, we compare NoisyGRPO with the baseline GRPO in terms of performance evolution, as well as the progression of the importance weight—defined as $\frac{\sigma_s^2}{\sigma_n^2 + \sigma_s^2}$ in Eq. 9—throughout training. We also report the average completion length produced by each method. From the results, we draw the following insights: (1) The importance weight decreases steadily throughout training, indicating that NoisyGRPO initially relies more on prior estimation driven by noise level, but gradually shifts to relying on correctness-based reward signals as training progresses. (2) NoisyGRPO consistently produces shorter completions than GRPO while achieving better performance, suggesting that its supervision mechanism over the CoT process encourages more concise and accurate reasoning steps. (3) Compared to GRPO, NoisyGRPO yields greater improvements on tasks requiring fine-grained visual understanding, such as Fine-grained Perception and Instance Reasoning, but performs slightly worse on knowledge-intensive tasks like Scene & Technology. This indicates that NoisyGRPO primarily enhances vision-language alignment in CoT reasoning, rather than boosting knowledge retrieval or external fact-based reasoning capabilities.

## 4.6   Ablation Study

**Method Design**  To investigate the contribution of each component in our proposed NoisyGRPO framework, we conduct an ablation study on the comprehensive MMStar [5] benchmark. As illustrated in Figure 4, we compare the performance trajectories of four variants throughout training. The baseline *GRPO* [44] corresponds to the original policy optimization algorithm introduced in DeepSeek-R1 [7]. The variant *GRPO with Noise Injection* incorporates our proposed noise-injected exploration policy on top of GRPO. *Naive NoisyGRPO* refers to a simplified version of NoisyGRPO, where the dynamic weighting mechanism between the observation and prior is removed—specifically, in Eq. 9, we set $\sigma_s^2 = \sigma_n^2$, thereby eliminating the adaptive fusion. From the results, we observe that naively injecting noise to perturb the visual inputs degrades policy learning performance, indicating that a well-designed mechanism is required to mitigate the adverse effects of noise. Furthermore, the NoisyGRPO with dynamic prior-observation fusion consistently outperforms other variants, demonstrating the effectiveness of this design in enhancing the reasoning capability of MLLMs during bootstrapping.

**Hyperparameter Sensitivity**  To assess the sensitivity of NoisyGRPO to its hyperparameters, we conduct experiments analyzing the impact of two key parameters in the Bayesian Advantage Estimation module. As defined in Eq. 8, $\alpha$ controls the confidence assigned to the observation, while $\gamma$, defined in Eq. 9, serves as a scale hyperparameter that modulates how strongly the variability of semantic rewards influences the prior uncertainty. As shown in Figure 4, NoisyGRPO exhibits relatively low sensitivity to $\alpha$ compared to $\gamma$, indicating that the bias introduced by the embedding-based reward function is smaller than that resulting from inaccurate prior estimation due to noise injection. We further extend the analysis to the reward-temperature parameter $\tau$. As shown in Table 3, NoisyGRPO maintains stable performance across a broad range of values for both hyperparameters on MMStar, AMBER, and MMERealworld benchmarks, consistently outperforming the vanilla GRPO baseline. These results confirm the robustness of NoisyGRPO and demonstrate that reasonable hyperparameter choices can be made without extensive tuning.

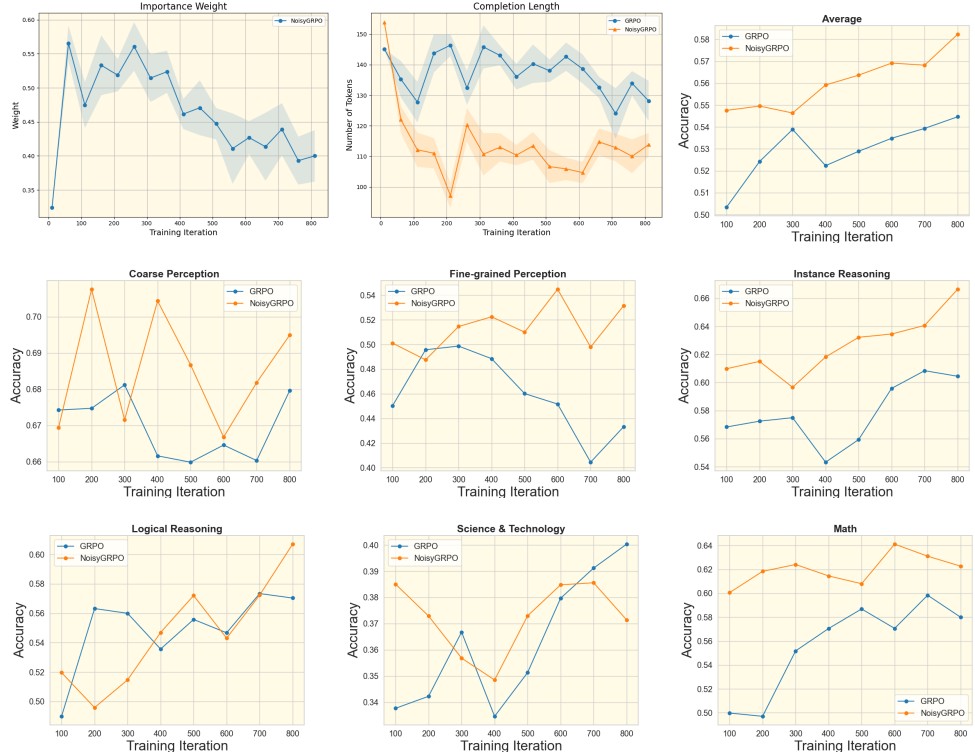

Figure 3: **Performance over iteration and training statistics.** We report the evaluation results of NoisyGRPO-3B on the MMStar benchmark to demonstrate the comprehensive capabilities of the MLLM. For both *Importance Weight* and *Completion Length*, the shaded regions represent the variance across samples.

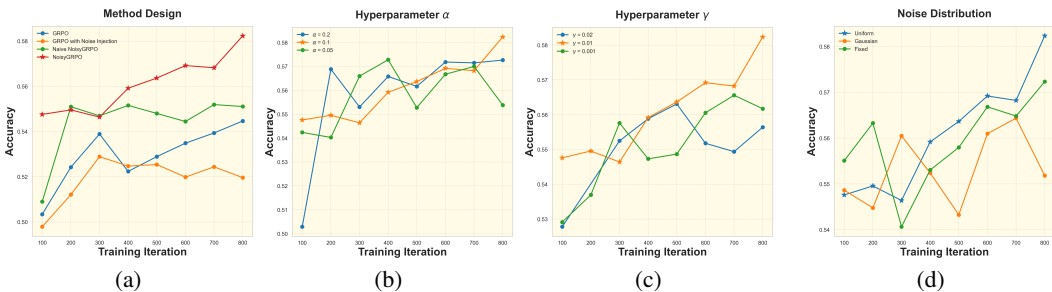

Figure 4: **Performance of Ablation Study.** We report the average performance on the MMStar benchmark across the following ablation settings: (a) Ablation of core design choices in NoisyGRPO; (b) Sensitivity to the hyperparameter $\alpha$ controlling observation confidence; (c) Sensitivity to the hyperparameter $\gamma$ modulating prior variance adaptation; (d) Comparison of different noise distributions used in the noise-injected exploration policy.

**Noise Distribution in Noise Injection** To investigate the effect of different noise distributions on policy learning in NoisyGRPO, we experiment with three types of injected noise during training. Specifically, we compare: (1) Uniform noise, which is the default choice in our final method; (2) Gaussian noise sampled from a normal distribution with mean 0 and variance 0.1; and (3) Fixed noise with a constant value of 0.5. As shown in Figure 4, we observe that noise sampled from a uniform distribution consistently leads to more stable and improved MLLM performance. In contrast, both Gaussian and fixed noise result in unstable training dynamics. We attribute this to the fact that uniform noise provides a broader and more balanced exploration space, while our proposed Bayesian

Table 3: **Sensitivity analysis of hyperparameters $\tau$.** Results under varying hyperparameter settings are reported on MMStar, AMBER, and MMERealworld benchmarks. The best performance for each metric is highlighted in bold. The configuration highlighted in blue corresponds to the setting used in the main results.

| | MMStar | | | | | | | AMBER | | | | | MMERW |
|---|---|---|---|---|---|---|---|---|---|---|---|---|---|
| | CP. | FP. | IR. | LR. | ST. | MA. | Avg. | Cs ↓ | Cov. ↑ | Hal. ↓ | Cog. ↓ | F1 ↑ | Acc |
| Vanilla GRPO | 68.0 | 43.3 | 60.4 | 57.0 | 40.0 | 58.0 | 54.5 | 6.7 | 68.5 | 44.6 | 4.2 | 89.2 | 40.8 |
| NoisyGRPO with different $\tau$ | | | | | | | | | | | | | |
| $\tau = 0.5$ | 66.1 | 50.9 | 62.5 | 56.2 | 37.5 | **65.9** | 56.5 | 6.7 | 64.3 | **37.5** | **2.5** | 89.5 | 43.0 |
| $\tau = 0.6$ | **69.5** | **53.2** | **66.6** | **60.7** | 37.1 | 62.3 | **58.2** | **6.6** | 67.7 | 44.3 | 3.4 | **90.3** | 44.0 |
| $\tau = 0.7$ | 69.1 | 52.3 | 63.4 | 58.2 | **39.3** | 56.2 | 56.4 | 7.4 | 65.8 | 42.4 | 3.0 | 90.0 | **44.7** |

Table 4: **Comparison of computational efficiency between GRPO and NoisyGRPO.** The reported numbers for both methods are tested on a single node with 8 A100 GPUs.

| Method | Wall-Clock Time | GPU-Hours | Peak GPU Memory |
|---|---|---|---|
| Qwen2.5-VL-3B | | | |
| GRPO | 6h12min | 49.6 | 57 GB |
| NoisyGRPO | 6h40min | 53.3 | 53 GB |
| Qwen2.5-VL-7B | | | |
| GRPO | 7h58min | 63.7 | 75 GB |
| NoisyGRPO | 7h52min | 62.9 | 77 GB |

Advantage Estimation effectively mitigates the adverse impact of high-variance noise during policy optimization.

**Training Efficiency Comparison**   To evaluate the practical scalability and efficiency of Noisy-GRPO, we measure its training cost and resource usage in comparison with GRPO [44]. Specifically, we report the wall-clock time, GPU-hours, and peak memory consumption for both methods under identical experimental settings. Here, NoisyGRPO and GRPO sample the same number of rollouts. As summarized in Table 4, the results show that NoisyGRPO maintains comparable efficiency to GRPO despite its additional sampling step, demonstrating its suitability for real-world deployment.

## 5   Limitation

While our proposed NoisyGRPO effectively enhances the Chain-of-Thought (CoT) reasoning ability of MLLMs, it also comes with certain limitations. Specifically, our method relies on the assumption that injecting Gaussian noise into the visual input perturbs the model's generation and that the degree of noise correlates with the correctness of the generated answer. This assumption may not hold in tasks such as coarse-grained perception, where detailed visual information is less critical. In such cases, the injected noise may fail to meaningfully affect the model's outputs, and the noise magnitude, used as a prior in Bayesian advantage estimation, can introduce significant bias. This can, in turn, negatively impact the learning of the policy in NoisyGRPO.

## 6   Conclusion

In this work, we present NoisyGRPO, a multimodal reinforcement learning framework designed to enhance the Chain-of-Thought reasoning capabilities of MLLMs. Without incurring additional computational cost, we introduce a noise injection strategy to improve policy exploration and address the limited diversity of generations in vanilla GRPO. To counteract the potential negative impact of injected noise, we propose a principled Bayesian Advantage Estimation method, where the injected noise level serves as a prior and the trajectory reward serves as the likelihood. This Bayesian modeling fuses both sources of information to compute a robust posterior estimate of trajectory advantage, enabling more accurate and robust policy optimization under visual uncertainty. Experimental results demonstrate the effectiveness of NoisyGRPO across diverse benchmarks, particularly under challenging RL settings with small-scale MLLMs.

# 7 Acknowledge

This work was supported by NSFC 62350610269, Shanghai Frontiers Science Center of Human-centered Artificial Intelligence, and MoE Key Lab of Intelligent Perception and Human-Machine Collaboration (ShanghaiTech University). This work was also supported by HPC platform of ShanghaiTech University.

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

# A  Preliminary

This section formalizes the reinforcement learning (RL) framework for post-training optimization of multimodal large language models (MLLMs). We introduce the mathematical foundation of Group Relative Policy Optimization (GRPO) [44], a state-of-the-art RL approach for language model alignment. Furthermore, we provide an overveiw of Bayesian Estimation and offer detailed derivations of the key equations introduced in the paper, starting from their original definitions.

## A.1  Group Relative Policy Optimization (GRPO)

GRPO [44] is a reinforcement learning (RL) algorithm specifically designed to enhance the reasoning capabilities of Large Language Models (LLMs). Unlike traditional approaches, GRPO eliminates the need for a value function and instead estimates advantages in a group-relative manner. Given a question-answer pair (q, a), the exploration policy $\pi_{\theta_{old}}$ samples a group of G individual responses $\{o_i\}_{i=1}^G$. The advantage of the $i$-th response is computed by normalizing the group-level rewards $\{r_i\}_{i=1}^G$:

$$\hat{A}_i = \text{GN}(r_i, \{r_j\}_{j=1}^G) = \frac{r_i - \text{mean}(\{r_j\}_{j=1}^G)}{\text{std}(\{r_j\}_{j=1}^G)} \tag{11}$$

where we define GN as the function for group normalization. GRPO adopts a clipped surrogate objective similar to PPO, with an additional KL divergence penalty to regularize the policy update:

$$J_{\text{GRPO}}(\theta) = \mathbb{E}_{(q,a)\sim\mathcal{D},\{o_i\}_{i=1}^G\sim\pi_{\theta_{\text{old}}}(\cdot|q)}$$

$$\left[ \frac{1}{G} \sum_{i=1}^G \min\left( \frac{\pi_\theta(o_i \mid q)}{\pi_{\theta_{\text{old}}}(o_i \mid q)} \hat{A}_i, \ \text{clip}(\frac{\pi_\theta(o_i \mid q)}{\pi_{\theta_{\text{old}}}(o_i \mid q)}, 1 - \epsilon, 1 + \epsilon) \ \hat{A}_i \right) - \beta D_{\text{KL}}(\pi_\theta \| \pi_{\text{ref}}) \right] \tag{12}$$

Here, $\pi_\theta$ and $\pi_{\theta_{\text{old}}}$ are the current and previous policies respectively, $\epsilon$ is the PPO clipping parameter, $\beta$ is the KL penalty coefficient, $\pi_{\text{ref}}$ is a reference policy, and $D_{\text{KL}}$ denotes KL divergence.

It is also worth noting that GRPO incorporates temperature-based sampling to promote exploration. The presence of both high- and low-quality responses within a group enables the group-relative advantage estimator to serve as a meaningful training signal. However, as noted in DAPO [58], GRPO suffers from a gradient vanishing issue when prompts yield identical reward values (e.g., all correct or all incorrect responses). In the main paper, we introduce NoisyGRPO, a set of enhancements designed to address this limitation and improve the robustness of GRPO.

## A.2  Bayesian Estimation

Bayesian estimation is a fundamental mechanism for probabilistic inference that refines a prior belief in light of new evidence. Given a prior distribution $p(\theta)$ over a latent variable $\theta$, and a likelihood function $p(y \mid \theta)$ representing the observed data $y$, the posterior distribution $p(\theta \mid y)$ is computed using Bayes' rule:

$$p(\theta \mid y) = \frac{p(y \mid \theta)p(\theta)}{p(y)} \tag{13}$$

Here, $p(y)$ is the marginal likelihood (or evidence), ensuring that the posterior is properly normalized.

Under the assumption that both the prior and the likelihood are Gaussian, and the latent variable $\theta$ is scalar:

$$p(\theta) = \mathcal{N}(\theta; \mu_0, \sigma_0^2), \quad p(y \mid \theta) = \mathcal{N}(y; \theta, \sigma_y^2), \tag{14}$$

the posterior $p(\theta \mid y)$ is also Gaussian. Its variance $\sigma_{\text{post}}^2$ and mean $\mu_{\text{post}}$ are given by:

$$\sigma_{\text{post}}^2 = \left( \frac{1}{\sigma_0^2} + \frac{1}{\sigma_y^2} \right)^{-1}, \qquad \mu_{\text{post}} = \left( \frac{\mu_0}{\sigma_0^2} + \frac{y}{\sigma_y^2} \right) \cdot \sigma_{\text{post}}^2 \tag{15}$$

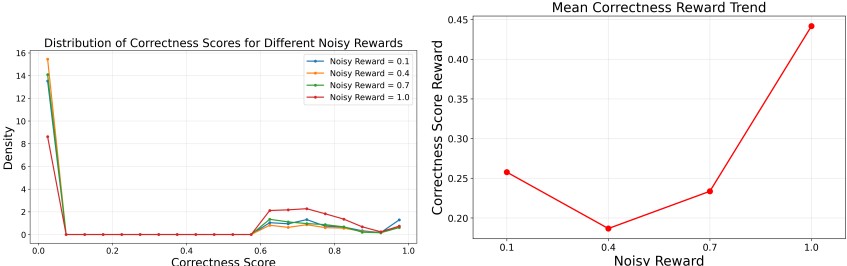

Figure 5: **Correlation of correctness and noise level.**

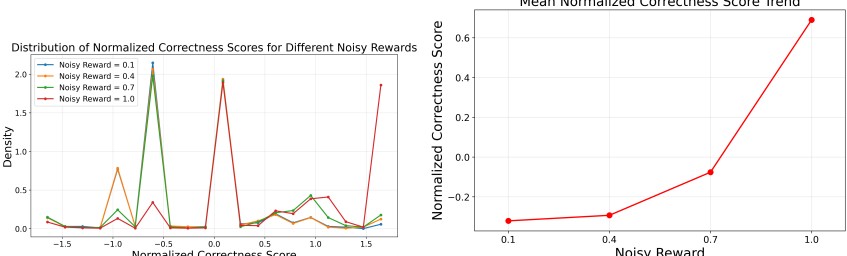

Figure 6: **Correlation of group normalized correctness and noise level.**

Alternatively, the posterior mean can be rewritten in the following equivalent form to highlight its interpolation behavior:

$$\mu_{\text{post}} = y + \frac{\sigma_y^2}{\sigma_0^2 + \sigma_y^2}(\mu_0 - y) \tag{16}$$

This form illustrates how the uncertainty and information from both the prior and the observation are combined with weighted contributions. The variances $\sigma_0^2$ and $\sigma_y^2$ play a key role in determining the influence of the prior and the observed data, representing the uncertainty in the prior estimate and the observation, respectively. While this interpolated form is convenient for interpretation, the expression in Eq. (4) is more commonly used in Bayesian analysis, as it explicitly shows the posterior as a precision-weighted average of the prior and observed values.

These insights into Bayesian updating inform our design of Bayesian advantage estimation in policy optimization, allowing us to better incorporate uncertainty from both the noise prior and the trajectory reward, as described in the main paper.

## B  Distribution Analysis of the Noise-level and Trajectory Reward

### B.1  Correlation between Prior and Observation

To demonstrate the rationality of our method design, we begin by analyzing the correlation between the noise level prior and the correctness reward(observation) during training. In Figure 5, we visualize the raw values of the noisy reward $r_i^n \in [0, 1]$ and the corresponding correctness scores $\in [0, 1]$. Specifically, we present both the distributions of their means and the overall numerical distributions. From these visualizations, we observe that when the noisy reward is below 0.7 (corresponding to cases where more than 30% of the steps in the image trajectory are corrupted with noise), the expected positive correlation between higher noisy reward and higher correctness score does not hold based on the raw values alone.

However, as NoisyGRPO applies group normalization from GRPO during advantage computation, we further provide the score distributions after group normalization. As illustrated in Figure 6, the distribution of the mean value of the correctness score after normalization exhibits a clear and consistent trend: the higher the noisy reward, the higher the correctness score. This shift indicates

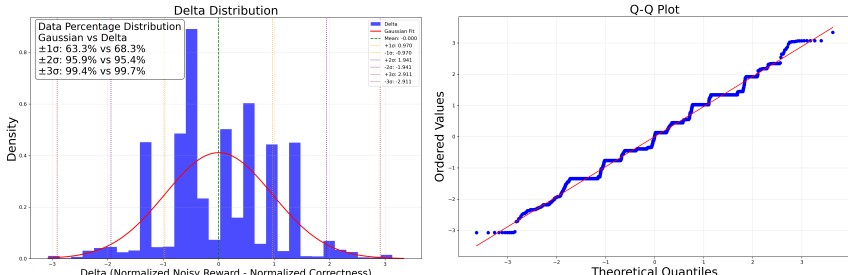

Figure 7: **Histogram and Q-Q plot of residual of observation and prior.** The Gaussian in red is the Gaussian distribution with identical mean and variance to the residual.

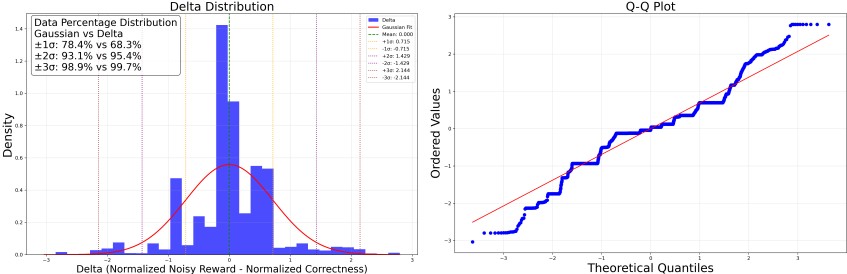

Figure 8: **Histogram and Q-Q plot of residual of posterior and observation.** The Gaussian in red is the Gaussian distribution with identical mean and variance to the residual.

that the inherent differences in sample characteristics and difficulty levels cause the effect of noise injection to vary significantly across samples. Consequently, group normalization is essential to our approach for stabilizing training.

## B.2 Verification of Gaussian Assumption

Our method is built upon a fundamental assumption regarding the underlying distribution of reward signals. Specifically, we assume that the prior estimation of quality follows a Gaussian distribution, i.e., $r_i \sim \mathcal{N}(\hat{r}_i^n, \sigma_n^2)$, where $\hat{r}_i^n$ is the estimated mean quality and $\sigma_n^2$ is the prior variance. Meanwhile, the observed semantic reward is modeled as a Gaussian observation of the true latent reward: $\hat{r}_i^s \sim \mathcal{N}(r_i, \sigma_s^2)$.

However, since the true latent reward $r_i$ is not accessible, we cannot directly validate this assumption. To address this, we employ indirect analysis and statistical testing to evaluate the plausibility of the Gaussian assumption underlying our method design. To this end, we construct two residual terms based on available quantities:

$$\epsilon_i^{(1)} = \hat{r}_i^s - \hat{r}_i^n \quad \text{(Observation vs. Prior)}$$

$$\epsilon_i^{(2)} = \hat{r}_i - \hat{r}_i^s \quad \text{(Posterior vs. Observation)}$$

Here, $\epsilon_i^{(1)}$ captures the deviation between the semantic observation and the prior estimation before fusion, while $\epsilon_i^{(2)}$ reflects the adjustment made by the posterior with respect to the observation. If both observation and prior are Gaussian as assumed, and the posterior is computed using standard Bayesian fusion, then both residuals should approximately follow zero-mean Gaussian distributions with variances that are analytically tractable. In the following, we empirically examine the distribution of these residuals to test the validity of our assumption.

**Residual of Observation and Prior** We first analyze the residual between the observation and prior, defined as the delta between the correctness score and the noisy reward. This quantity reflects how much the observed semantic signal deviates from the prior belief before any posterior fusion.

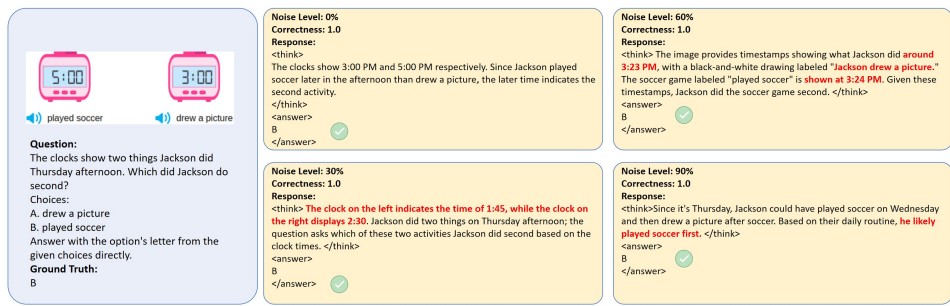

Figure 9: **Illustration of "Answer Correctness is a Partial Observation"**

To assess the distributional characteristics of this residual, we visualize it in Figure 7 using both a histogram and a Quantile-Quantile (Q-Q) plot against a standard Gaussian distribution. Additionally, we conduct formal normality tests to statistically evaluate the Gaussianity of the residual. The results are as follows:

- **Shapiro-Wilk Test**: $W = 0.983$, $p$-value $= 7.416 \times 10^{-24}$. The Shapiro-Wilk test evaluates whether a sample comes from a normally distributed population. A $W$ value close to 1 indicates that the data is close to normal; the low p-value here, due to the large sample size, suggests statistical significance but does not necessarily imply a practically meaningful deviation from normality.

- **Kolmogorov-Smirnov Test**: $D = 0.110$, $p$-value $= 2.432 \times 10^{-52}$. The Kolmogorov-Smirnov (K-S) test compares the empirical distribution of the sample to a reference Gaussian distribution. The statistic $D$ measures the maximum distance between the two cumulative distribution functions. Again, while the small p-value indicates statistical rejection of normality, the K-S test is known to be sensitive in large samples, and the shape shown in the Q-Q plot remains approximately Gaussian.

As can be observed from both the visualizations and the statistical test results, the residual approximately follows a Gaussian distribution. Although the extremely low p-values may suggest deviations from strict normality due to the large sample size, the overall shape and symmetry of the distribution, as shown in the Q-Q plot support the plausibility of the Gaussian assumption for this residual term.

**Residual of Posterior and Observation**   In this section, we analyze the distribution of the residual between the posterior and the observation, which corresponds to the difference between the posterior advantage and the correctness score. Similar to the previous analysis, we present both a histogram and a Quantile-Quantile (Q-Q) plot in Figure 8 to visualize the shape of the residual distribution.

- **Shapiro-Wilk Test**: $W = 0.932$, $p$-value $= 1.694 \times 10^{-42}$
  The Shapiro-Wilk test assesses the null hypothesis that the sample is drawn from a normal distribution. Although the p-value is very small (due to the large sample size), the $W$ value remains reasonably high, suggesting the distribution retains approximate Gaussian characteristics.

- **Kolmogorov-Smirnov Test**: $D = 0.190$, $p$-value $= 3.529 \times 10^{-155}$
  The Kolmogorov-Smirnov test measures the maximal deviation $D$ between the empirical distribution and a reference Gaussian. Despite the low p-value (again, expected in large-scale settings), the moderate $D$ value supports the plausibility of a Gaussian assumption.

Taken together with the results from the prior-observation residual analysis, these findings provide empirical support for the validity of our Gaussian assumption in modeling both the prior and the observation noise.

## C   Visualization

**Answer Correctness is a Partial Observation**   One of our key motivations stems from the fact that the correctness of the final answer cannot fully reflect the overall rollout quality, especially in

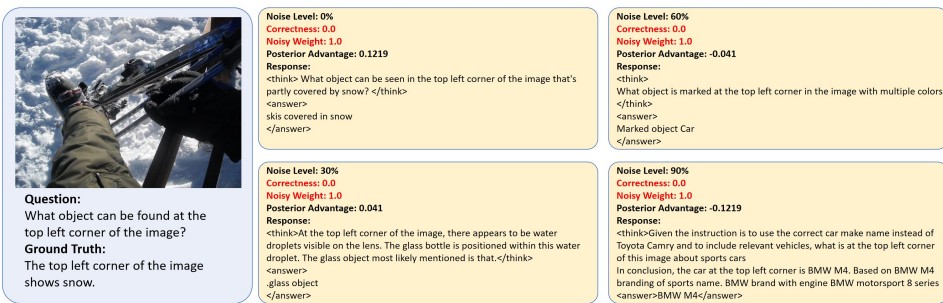

Figure 10: **Illustration of Training rollouts and trajectory reward.**

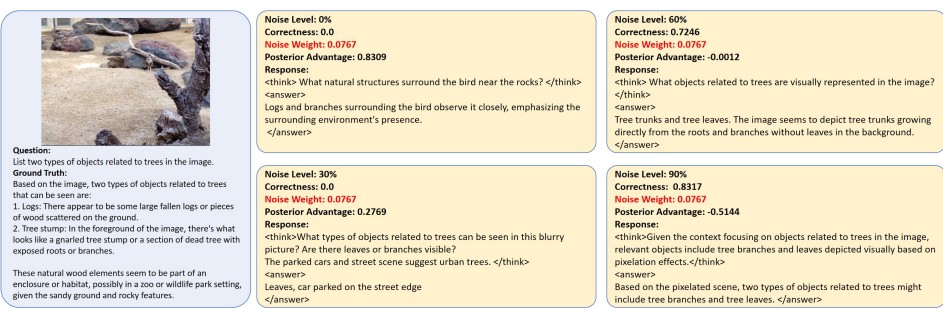

Figure 11: **Illustration of Training rollouts and trajectory reward.**

complex multi-step reasoning settings like Chain-of-Thought (CoT). To qualitatively illustrate this phenomenon, we provide a concrete example in Figure 9. In this example, NoisyGRPO samples multiple rollouts under different noise levels. Interestingly, all sampled answers are correct with respect to the final answer, yet the quality of the intermediate CoT reasoning varies noticeably. This demonstrates that relying solely on answer correctness as a training or evaluation signal may overlook significant variations in reasoning quality. Hence, a more nuanced reward structure, as introduced in our method, is necessary to capture the semantics of the full reasoning trajectory.

**Weight in Bayesian Estimation**    In this section, we qualitatively illustrate how the training process dynamically balances the weights between the prior and observation in our Bayesian estimation framework. Specifically, the weight assigned to the observation is defined as

$$\frac{\sigma_s^2}{\sigma_n^2 + \sigma_s^2},$$

where $\sigma_n^2$ and $\sigma_s^2$ denote the variances of the noise prior and semantic observation, respectively.

We first demonstrate an extreme case where all the answers in the sampled trajectories are incorrect. In this situation, vanilla GRPO fails to provide meaningful training signals, as the semantic correctness reward becomes uniformly zero. However, as shown in Figure 10, NoisyGRPO adaptively reduces the weight of the observation (correctness score) to zero, allowing the noisy prior alone to contribute to the training signal. This enables the policy model to continue learning from the noise level's implication on CoT quality, even in the absence of correct final answers.

In contrast, when the correctness scores exhibit high variance—meaning some answers are correct and others are not, as illustrated in Figure 11—NoisyGRPO assigns a balanced weight between the prior and the observation. This allows the model to leverage both the semantic correctness and the injected noise level to assess the overall trajectory quality more accurately.

These examples highlight the importance of dynamic weighting in our Bayesian estimation, enabling stable and informative policy updates across varying rollout qualities.

**Rollouts Sampling Comparison with Vanilla GRPO**    In this section, we compare the policy exploration characteristics of vanilla GRPO and our proposed NoisyGRPO during rollout sampling.

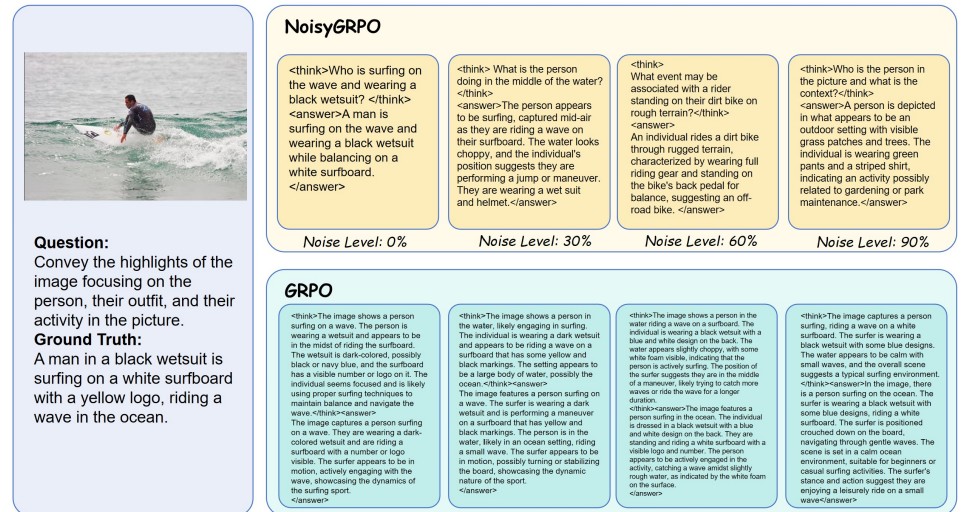

Figure 12: **Comparison of rollouts generation between vanilla GRPO and NoisyGRPO.**

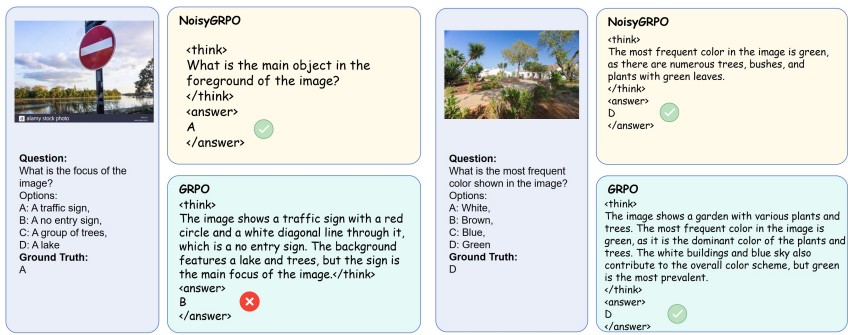

Figure 13: **CoT Inference comparison between GRPO and Noisy.**

Specifically, we examine the differences between two exploration strategies: one that solely relies on temperature sampling to generate multiple rollouts, and another that introduces noise injection to the inputs to diversify the rollouts.

As illustrated in Figure 12, we observe that rollouts sampled by vanilla GRPO tend to exhibit high similarity to each other. This limited diversity constrains the scope of policy exploration and may lead to suboptimal learning. In contrast, NoisyGRPO, by injecting structured noise during rollout generation, encourages exploration of more diverse reasoning paths and CoT structures. This diversity allows the model to better capture the relationship between reasoning quality and task performance, ultimately leading to more robust policy improvement.

**Inference Comparison with Vanilla GRPO** In this section, we present a qualitative visualization comparing the CoT inference behavior of vanilla GRPO and our proposed NoisyGRPO. As shown in Figure 13, we observe that rollouts generated by NoisyGRPO tend to have shorter chains of thought while maintaining comparable answer accuracy. This suggests that NoisyGRPO achieves more efficient CoT reasoning by avoiding unnecessary elaboration.

Moreover, we find that in some cases, NoisyGRPO generates an intermediate question within the CoT, which serves as a refinement or decomposition of the original input question. This strategy allows the model to clarify its internal reasoning path and arrive at a more accurate final answer. In contrast, vanilla GRPO often produces longer, redundant reasoning steps that do not necessarily contribute to better accuracy.

These observations indicate that incorporating noisy priors during training encourages the policy model to develop more concise and targeted reasoning strategies during inference.

**Comparison with Rule-Based Reward**  To validate our choice of using the text embedding model SBERT as the reward model, we compare it with a popular rule-based reward approach. Specifically, we adopt the average F1 score of ROUGE-1, ROUGE-2, and ROUGE-L as the reward. The results are summarized in Table 5. As shown, SBERT consistently outperforms the rule-based reward in our experimental setting, likely due to its ability to capture semantic similarity beyond exact token overlap, which is especially important for evaluating open-ended answers where lexical variance is common.

Table 5: Comparison of SBERT-based reward vs. rule-based ROUGE reward on MMStar, AMBER, and MMERealworld metrics.

| Method | MMStar | | | | | | | AMBER | | | | | MMERW |
|---|---|---|---|---|---|---|---|---|---|---|---|---|---|
| | CP. | FP. | IR. | LR. | ST. | MA. | Avg. | Cs $\downarrow$ | Cov. $\uparrow$ | Hal. $\downarrow$ | Cog. $\downarrow$ | F1 $\uparrow$ | Acc |
| Rouge reward | 64.9 | 49.5 | 64.8 | 51.3 | 38.0 | 52.6 | 53.5 | 8.0 | 66.4 | 46.5 | 3.4 | 89.3 | 43.6 |
| SBERT reward | 69.5 | 53.2 | 66.6 | 60.7 | 37.1 | 62.3 | 58.2 | 6.6 | 67.7 | 44.3 | 3.4 | 90.3 | 44.0 |

# D   Scalability Analysis

To evaluate the scalability of NoisyGRPO with respect to training data size, we conduct experiments using subsampled datasets containing 3k and 6k examples, compared to the original 13k dataset. Specifically, we randomly sample 3k and 6k subsets from the full training corpus and train both GRPO and NoisyGRPO under identical training steps to ensure a fair comparison.

As shown in Table 6, NoisyGRPO maintains consistent gains over the GRPO baseline across all dataset sizes, demonstrating its robustness and scalability when exposed to larger data volumes. Furthermore, performance improvements are observed as the amount of training data increases, confirming that NoisyGRPO effectively leverages additional data to enhance policy learning. These results highlight that NoisyGRPO is a scalable and data-efficient optimization framework, capable of leveraging additional training samples to further enhance multimodal reasoning performance without loss of stability.

Table 6: Scalability analysis of NoisyGRPO. Performance comparison between GRPO and Noisy-GRPO trained with subsampled datasets (3k and 6k samples). NoisyGRPO consistently outperforms the baseline under all data scales, and its performance improves with larger datasets. Best results in each column are highlighted.

| Method / Size | MMStar | | | | | | | AMBER | | | | | MMERW |
|---|---|---|---|---|---|---|---|---|---|---|---|---|---|
| | CP. | FP. | IR. | LR. | ST. | MA. | Avg. | Cs ↓ | Cov. ↑ | Hal. ↓ | Cog. ↓ | F1 ↑ | Acc |
| GRPO 3k | 66.4 | 44.6 | 59.7 | 53.8 | 35.1 | 56.7 | 52.7 | 7.1 | 66.1 | 46.9 | 3.8 | 88.9 | 38.9 |
| NoisyGRPO 3k | 66.7 | 51.8 | 61.2 | 56.4 | 35.3 | 62.3 | 55.6 | 6.7 | 63.5 | 36.8 | 2.2 | 89.9 | 44.5 |
| GRPO 6k | 67.2 | 45.6 | 58.7 | 53.2 | 40.2 | 54.1 | 53.2 | 7.0 | 67.3 | 48.8 | 4.7 | 88.3 | 38.1 |
| NoisyGRPO 6k | 69.1 | 52.2 | 64.3 | 54.7 | 39.7 | 59.5 | 56.6 | 6.4 | 65.4 | 40.2 | 3.1 | 90.1 | 44.1 |

