# OpenReview forum: "NoisyGRPO: Incentivizing Multimodal CoT Reasoning via Noise Injection and Bayesian Estimation"
_NeurIPS.cc/2025/Conference — NeurIPS 2025 poster_

### Official Review · Reviewer_TGQ6 · 2025-06-17

**Clarity:** 3
**Significance:** 3
**Originality:** 3
**Rating:** 4
**Confidence:** 4

**Summary:**

NoisyGRPO tackles the limited generalization of existing methods for MLLMs by injecting controllable Gaussian noise into input images during rollouts, thereby widening exploration and providing a trajectory-level signal of reasoning difficulty. To prevent this noise from destabilizing learning, the framework casts advantage estimation as a Bayesian update that fuses a noise-based prior with the observed reward, yielding a posterior that adaptively weights exploration versus correctness. Both components are lightweight. Across different benchmarks, it consistently outperforms vanilla GRPO, while bringing the most pronounced gains to smaller models. Ablations show that uniform noise combined with the dynamic Bayesian fusion drives these improvements, though the method can underperform on coarse-perception tasks where noise fails to perturb visual cues, highlighting a key limitation meaningfully.

**Questions:**

See Weaknesses above.

**Ethical Concerns:**

["NO or VERY MINOR ethics concerns only"]

**Final Justification:**

The authors addressed most of my questions and concerns. Considering that the overall quality and the novelty of the work, I recommend a rating of 4: Borderline Accept.

**Limitations:**

yes

**Quality:**

3

**Strengths And Weaknesses:**

Strengths:
1. NoisyGRPO perturbs each input image with controllable Gaussian noise during rollout collection, significantly broadening the search space and promoting visually grounded CoT trajectories.
2. A Bayesian advantage estimator treats the noise level as a prior and the trajectory reward as the likelihood, enabling dynamic down-weighting of noisy or unreliable rollouts and preventing learning collapse.
3. NoisyGRPO achieves improvements on benchmarks while remaining lightweight.

Weaknesses:
1. The method promotes grounded answers but at the expense of descriptive breadth. Its captions score lower on AMBER Cov. metric, which penalizes omissions of image elements.
2. On certain models, vanilla GRPO produces more concise CoT with comparable accuracy.
3. Ablations show that naïve noise injection or sub-optimal γ in the Bayesian fusion can degrade performance, indicating that the gains are sensitive and depend on careful tuning.
4. The accuracy reward relies on SBERT similarity, and all hyperparameters are selected directly on MMStar without a held-out validation set, raising concerns about embedding bias and benchmark overfitting.

---

> ### Author Rebuttal · Authors · 2025-07-30
>
> **We thank the reviewer for highlighting the strengths of our method, including the noise-driven exploration, Bayesian estimator, and lightweight design. We address the reviewer’s concerns below.**
>
> ---
> **W1:** *Promote grounded answers but at the expense of descriptive breadth*
>
> ---
>
> We argue that **AMBER Coverage has inherent limitations** and does **not fully support the claim** that our method promotes grounding at the expense of descriptive breadth. In AMBER's generative setting, each case has on average **5.5 ground truth captions**, and our method shows at most a **2.7% decrease in Coverage** compared to the strongest baseline—equivalent to merely **0.14 missing words per case**, which is a negligible trade-off considering the benefit of reduced hallucination.
>
> Moreover, we observe that **many hallucination-mitigating methods exhibit similar behavior** on AMBER, where improving factual accuracy can naturally lead to slightly lower coverage scores. This suggests that the metric may not fully capture the quality of grounded yet concise descriptions. To illustrate this, we propose a **CHR** to measure how much the hallucination rate decreases relative to the drop in coverage. A higher value indicates more effective hallucination reduction with less sacrifice in coverage.
>
> $\text{CHR} = \frac{|\Delta \text{Hallucination Rate}|}{|\Delta \text{Coverage}|}$
>
> This trend is not unique to our method — as the table shows, several recent strong baselines also experience similar trade-offs between hallucination reduction and coverage."
>
> | Method         | Cs ↓ | Cov. ↑ | Hal. ↓ | Cog. ↓ | CHR↑  |
> |----------------|-------|--------|--------|--------|------|
> | LLaVA-1.5 7B   | 7.8   | 51.0   | 36.4   | 4.2    | -    |
> | HA-DPO[1]        | 7.2   | 33.6   | 19.7   | 2.6    | 0.96 |
> | CLIP-DPO[2]       | 3.7   | 47.8   | 16.6   | 1.3    | 6.19 |
> | Qwen2.5-vl-3B  | 7.6   | 69.9   | 55.7   | 5.8    | -    |
> | NoisyGRPO 3B   | 6.6   | 67.7   | 44.3   | 3.4    | 5.18 |
> | Qwen2.5-vl-7B  | 4.8   | 63.6   | 27.5   | 1.6    | -    |
> | NoisyGRPO 7B   | 4.2   | 63.2   | 23.9   | 1.2    | **9.00** |
>
> From the table, we can see that other methods on AMBER also exhibit a decrease in hallucination accompanied by a drop in coverage, while NoisyGRPO 7B achieves the best CHR score. This suggests that **NoisGRPO reduces hallucination more effectively than others**, while incurring a relatively smaller drop in coverage, as reflected by the highest CHR score among all methods.
>
>
> Reference:
>
> [1] Beyond Hallucinations: Enhancing LVLMs through Hallucination-Aware Direct Preference Optimization, arXiv 2023, cited 148 times.
>
> [2] CLIP-DPO: Vision-Language Models as a Source of Preference for Fixing Hallucinations in LVLMs, ECCV 2024, cited 26 times.
>
> ---
> **W2:** *Vanilla GRPO produces more concise CoT with comparable accuracy*
>
> ---
>
> We acknowledge this observation. However, we would like to clarify that the results on the MME-CoT efficiency category **do not fully support the claim that vanilla GRPO produces more concise CoT**. First, MME-CoT evaluates CoT quality based on **relevance and reflection**, which do not comprehensively measure consciousness. Furthermore, we conducted a quantitative analysis of the **actual CoT lengths during inference**. As shown in the table below, NoisyGRPO consistently produces shorter CoTs across benchmarks:
>
> | CoT Length    | MMStar  | AMBER  | MM-RealWorld |
> | ------------- | ------- | ------ | ------------ |
> | GRPO          | 295     | 335    | 358                |
> | **NoisyGRPO** | **172** | **53** | **236**      |
>
> We observe that NoisyGRPO generates significantly shorter CoTs compared to vanilla GRPO, while achieving better performance on most benchmarks. **From the perspective of CoT length as a proxy for conciseness**, we argue that **NoisyGRPO demonstrates a clear advantage in generating more concise CoTs.**
>
> Moreover, we provide justification for the shorter CoT length on AMBER: since all questions are fixed as "Describe the image," NoisyGRPO typically reframes the task by posing a question about the image's main subject. This behavior is illustrated in the visualization in Figure 10 of the appendix.
>
> ---
> **W3:** *The gains are sensitive and depend on careful tuning.*
>
> ---
>
> We would like to clarify that we did **not perform extensive hyperparameter tuning**, and we humbly argue that introducing new hyperparameters is often **inevitable** when proposing a new framework. To demonstrate the **robustness of our method to hyperparameter choices**, we conduct a sensitivity analysis of $\gamma$, as presented in the table below.
>
> |       | MMStar           |       |       |       |       |       |       | AMBER          |       |       |       |       | MMERealworld |
> |---------|------------------|-------|-------|-------|-------|-------|-------|----------------|-------|-------|-------|-------|--------------|
> |         | CP.  | FP.  | IR.  | LR.  | ST.  | MA.  | Avg.  | Cs ↓ | Cov. ↑ | Hal. ↓ | Cog. ↓ | F1 ↑  | Acc          |
> | vanilla GRPO | 68.0  | 43.3 |  60.4 |  57.0  | 40.0  | 58.0  | 54.5  | 6.7  | 68.5  | 44.6  | 4.2 |  89.2 |  40.8 |
> | NoisyGRPO with $\gamma$ = 0.015   | 66.0  | 54.1 | 63.0 | 52.4 | 36.2 | 59.1 | 55.2  | 6.8   | 69.8   | 45.2   | 2.9    | 89.3  | 44.7         |
> |NoisyGRPO with  $\gamma$ = 0.01    | 69.5  | 53.2 | 66.6 | 60.7 | 37.1 | 62.3 | 58.2  | 6.6   | 67.7   | 44.3   | 3.4    | 90.3  | 44.0         |
> | NoisyGRPO with $\gamma$ = 0.005   | 67.1  | 48.9 | 62.8 | 57.9 | 37.0 | 61.7 | 56.9  | 6.4   | 65.2   | 38.4   | 2.6    | 90.5  | 43.2         |
>
> The results show that NoisyGRPO consistently outperforms the baseline across a small but meaningful range of $\gamma$ values, **demonstrating the method’s robustness and indicating that its performance gains do not rely on fine-grained hyperparameter tuning.**
>
> ---
> **W4:** *Concerns about embedding bias and benchmark overfitting.*
>
> ---
>
> **Embedding bias**
>
> We acknowledge the reviewer’s concern that SBERT may introduce bias as a reward function. This is indeed a **key consideration in our method design**. To mitigate this issue, our Bayesian advantage estimation explicitly incorporates a prior term that reflects the expected trajectory quality based on noise levels. This mechanism **dynamically balances** the SBERT-based semantic reward and the prior estimation, thereby **reducing over-reliance on the potentially biased SBERT signal** during training.
>
> Moreover, as detailed in our response to *reviewer nP6a*, Weakness 4, we conducted ablation studies using a rule-based reward function. The results consistently show that **SBERT outperforms the rule-based alternatives** in our experiments, further supporting its utility despite its known limitations.
>
> **Benchmark Overfitting**
>
> As addressed in our response to previous W3, the parameter settings optimized on MMStar are **not optimal but still yield strong results for other benchmarks**. For example, $\gamma = 0.015$ achieves the best performance on MMERealworld, while $\gamma = 0.01$ performs best on AMBER. Nonetheless, our method **achieves comparable and strong performance** across these benchmarks, which demonstrates that **NoisyGRPO does not overfit to MMStar**.
>
> ---
> **We sincerely welcome the opportunity to further discuss our work during the discussion phase and thank the reviewers for their valuable feedback.**

---

> > ### Comment · Reviewer_TGQ6 · 2025-08-02
> > **Official Comment by Reviewer TGQ6**
> >
> > Thank you for responding in depth to all of my questions/concerns. Please add these additional results, especially the results from robustness experiments, into the paper, space permitting.

---

> > > ### Author Response · Authors · 2025-08-03
> > >
> > > We sincerely appreciate the reviewer's constructive feedback and recognition of our work. Thank you for your valuable suggestions regarding the robustness experiments. We will incorporate these additional results into the revised manuscript to further strengthen the paper.
> > >
> > > Your insightful comments have greatly helped us improve the quality of this work. Please let us know if there are any other points that need clarification or modification.

---

### Official Review · Reviewer_RksJ · 2025-07-02

**Clarity:** 2
**Significance:** 3
**Originality:** 2
**Rating:** 4
**Confidence:** 3

**Summary:**

This paper proposes NoisyGRPO, an extension of Group Relative Policy Optimisation (GRPO) that targets general chain of thought reasoning in multimodal LLMs. The authors combine two ideas, 1. Noise injected exploration to widen the exploration space and implicitly labels “easy” (clean) versus “hard” (noisy) trajectories. 2. Bayesian advantage estimation, a closed form Gaussian update to yield  a posterior that down-weights noisy but correct trajectories and guides the PPO style policy update. Experiments fine-tuned on  Qwen-2.5-VL compare against vanilla GRPO and show gains and some ablations on contributions from both factors are presented.

**Questions:**

- Comparison to DPO & R1: Can you add a run of DPO or VLM-R1 on the same data for context?

- Text-only generalisation, if the visual stream is empty (pure language tasks), does NoisyGRPO degrade to GRPO or hurt?

- wondering what the relative wall-clock vs. vanilla GRPO is when using equal numbers of rollouts per iteration?

- Have you tried how performance scales if the 13 k VQA set is subsampled to e.g. 3 k or 6 k?

Looking forward to the discussion in rebuttal, any thoughts on the questions or limitations would be nice.

**Ethical Concerns:**

["NO or VERY MINOR ethics concerns only"]

**Final Justification:**

Some of the additional results and clarifications make the paper stronger.

**Limitations:**

> Dataset size and domain scope are small, results may not transfer to web-scale pre-training or to other modality reasoning.

> Baseline coverage limited, no uncertainty estimates (std / CI) reported for main numbers (minor point).

> Method currently addresses exploration, not reward sparsity, might fail if answers are long-form or require external tools

**Quality:**

2

**Strengths And Weaknesses:**

### Strengths

- Conceptual novelty that turns  “add noise for robustness” trick into a principled prior inside GRPO, an elegant closed-form update I would say.
- Consistent gains on three benchmark families, especially for smaller 3B models. I personally liked  seeing gains on the smaller regime.
- Ablations separate noise-only, naïve fusion, and dynamic fusion, also include a parameter study.
- There is some disclosure of computational costs involved during training time.
- Since the nose only during roll-out, it's a sign of being cost-neutral.

### Weakness

- Prior assumes a monotonic relation between noise and difficulty that essentially breaks down for tasks where coarse cues suffice (acknowledged by authors).
- Training corpus is tiny (13 k VQA pairs) and settings are  a bit narrow, no study on larger, diverse datasets or on language-only tasks.
- Usual baselines for RL are not highlighted e.g. DPO, vanilla rejection sampling, andit  would be nice to include one value-based method.
- Computation cost is still heavy for academic or other practitioners, a minor limitation, I would say!
- Readability of Figure-2 and Table-2 can be better!
- Somehow, the presentation feels rushed. It can be improved.

---

> ### Author Rebuttal · Authors · 2025-07-29
>
> **We sincerely thank the reviewer for recognizing the conceptual novelty of introducing a principled noise-based prior in GRPO, the consistent performance improvements, the thorough ablation studies, and the transparent reporting of computational costs. We appreciate your encouraging feedback and address the remaining concerns below.**
>
> ---
> **Q1 & W3:** *Comparison to DPO & R1*
>
> ---
>
> Thank you for the great suggestion.
> First, we clarify that VLM-R1, when applied to our VQA training data, does not introduce additional design beyond the GRPO algorithm and thus is the **same as our GRPO baseline**. For DPO, we empirically demonstrate that the **NoisyGRPO outperforms DPO baseline**. Specifically, we extracted the same number of training samples from MMRLHF and trained a DPO model using the recommended settings from LLaMA Factory. The results are summarized in the table below.
>
> | Method      | CP. | FP. | IR. | LR. | ST. | MA. | Avg. | Cs ↓ | Cov. ↑ | Hal. ↓ | Cog. ↓ | F1 ↑ | Acc |
> |-------------|-----|-----|-----|-----|-----|-----|------|------|--------|---------|---------|------|------|
> | **DPO**     | 61.6 | 46.3 | 56.2 | 54.1 | 34.0 | 52.2 | 50.7 | 7.8  | 70.6   | 49.2   | 4.4   | 87.3 | 41.0 |
> | **NoisyGRPO** | 69.5 | 53.2 | 66.6 | 60.7 | 37.1 | 62.3 | 58.2 | 6.6  | 67.7   | 44.3   | 3.4   | 90.3 | 44.0 |
>
> From the results, we observe a **significant performance gap between the DPO baseline and NoisyGRPO** across multiple metrics. This discrepancy can be attributed to the higher data efficiency of NoisyGRPO, whereas DPO requires a larger amount of high-quality preference data to achieve competitive performance.
>
> ---
> **Q2 & W2:** *Text-only generalisation*
>
> ---
> Although text-only generalization is beyond the current scope of our work, we conducted exploratory evaluations on MMLU and found that **NoisyGRPO exhibits potential for generalization to the text domain.**
>
> Specifically, we compared GRPO, NoisyGRPO, and the pretrained MLLM.
>
> Because the RL training prompts explicitly require the model to perform Chain-of-Thought reasoning grounded on visual input, multimodal models often fail to follow instructions properly when images are absent. This results in a **high format error rate** (exceeding 60%) for both GRPO and NoisyGRPO on text-only samples. To mitigate this, we **only report performance on samples where the model produces valid (format-correct) answers**.
>
> | Model                          | Other | STEM | Humanities | Social & Sciences | Overall Acc | Format Error Rate |
> |-------------------------------|:-----:|:----:|:----------:|:-----------------:|:-----------:|:-----------------:|
> | Qwen2.5-VL-3B                 | 63.8  | 58.0 |    55.8    |       72.1        |    61.6     |         -         |
> | Qwen2.5-VL-7B                 | 74.8  | 66.8 |    60.1    |       79.1        |    69.0     |         -         |
> | **CoT Reasoning with Qwen2.5-VL-3B** ||||| |
> | GRPO                          | 73.9  | 62.0 |    53.3    |       73.2        |    66.4     |       77.3        |
> | NoisyGRPO                     | 71.1  | 56.9 |    68.1    |       74.3        |    68.9     |       85.5        |
> | **CoT Reasoning with Qwen2.5-VL-7B** ||||| |
> | GRPO                          | 76.3  | 70.4 |    63.0    |       79.1        |    72.6     |       61.8        |
> | NoisyGRPO                     | 83.3  | 81.0 |    68.8    |       82.5        |    79.8     |       75.3        |
>
> We observe that, despite the high format error rates caused by the absence of visual input, **NoisyGRPO consistently outperforms GRPO on valid outputs across most categories**. This suggests that our method has potential for generalization beyond vision-language tasks to purely text-based scenarios.
>
> We acknowledge that these results are preliminary, and we plan to conduct more extensive evaluations to better understand and improve text-only generalization in future work.
>
> ---
> **Q3 & W4:** *Statistics for Training*
>
> ---
>
> Thank you for the insightful question. We clarify that NoisyGRPO and GRPO sample the **same number of rollouts per iteration**, making their computation cost directly comparable.
>
> The table below summarizes wall-clock time, GPU-hours, and peak memory usage measured on a single node with 8 A100 GPUs:
>
> | Model              | Method     | Wall-Clock Time | GPU-Hours      | Memory Usage     |
> |-------------------|------------|------------------|----------------|------------------|
> | Qwen2.5-VL-3B      | GRPO       | 6h12min          | 49.6 A100 hours | Peak GPU 57GB    |
> |                   | NoisyGRPO  | 6h40min          | 53.3 A100 hours | Peak GPU 53GB    |
> | Qwen2.5-VL-7B      | GRPO       | 7h58min          | 63.7 A100 hours | Peak GPU 75GB    |
> |                   | NoisyGRPO  | 7h52min          | 62.9 A100 hours | Peak GPU 77GB    |
>
> We acknowledge that the overall computation cost remains substantial, which is a common challenge in large-scale vision-language RL research. The 13K dataset size already pushes our computational budget limits; with additional resources, we plan to explore even larger datasets to further validate our approach.
>
> ---
> **Q4:** *Training with Fewer Data*
>
> ---
>
> Great suggestion! We conduct experiments with subsampled data (e.g., 3k and 6k) and observe that NoisyGRPO maintains **consistent gains** over baselines, demonstrating its **scalability** in leveraging more training data to improve performance.
>
> Specifically, we sampled 3k and 6k training sets from the original 13k dataset and conducted experiments. To ensure a fair comparison, we set the number of training steps for the 3k and 6k subsets to be the same as that used for training on the full 13k dataset. The results are shown in the table below:
>
> | Method                | MMStar           |       |       |       |       |       |       | AMBER          |       |       |       |       | MMERealworld |
> |-----------------------|------------------|-------|-------|-------|-------|-------|-------|----------------|-------|-------|-------|-------|--------------|
> |                       | CP.  | FP.  | IR.  | LR.  | ST.  | MA.  | Avg.  | Cs ↓ | Cov. ↑ | Hal. ↓ | Cog. ↓ | F1 ↑  | Acc          |
> | GRPO 3k     | 66.4 | 44.6 | 59.7 | 53.8 | 35.1 | 56.7 | 52.7  | 7.1  | 66.1   | 46.9   | 3.8    | 88.9  | 38.9         |
> | NoisyGRPO 3k | 66.7 | 51.8 | 61.2 | 56.4 | 35.3 | 62.3 | 55.6  | 6.7  | 63.5   | 36.8   | 2.2    | 89.9  | 44.5         |
> | GRPO 6k     | 67.2 | 45.6 | 58.7 | 53.2 | 40.2 | 54.1 | 53.2  | 7.0  | 67.3   | 48.8   | 4.7    | 88.3  | 38.1         |
> | NoisyGRPO 6k | 69.1 | 52.2 | 64.3 | 54.7 | 39.7 | 59.5 | 56.6  | 6.4  | 65.4   | 40.2   | 3.1    | 90.1  | 44.1         |
>
>
> We observe from the results above that NoisyGRPO’s performance improves as the training data size increases. Moreover, at each data scale, NoisyGRPO consistently outperforms GRPO. Therefore, we consider NoisyGRPO to be a scalable approach.
>
> ---
>
> **W4:** *Prior assumes a monotonic relation between noise and difficulty*
>
> ---
>
> We would like to provide some additional clarification on this limitation. We believe this **assumption holds in most current vision-language scenarios**, as evidenced by the results in Table 2. For cases where the assumption may fail, such as coarse perception tasks, we observe that this failure does **not cause a significant performance drop** in NoisyGRPO — it achieves results comparable to the baseline (e.g., less than 1% decrease on coarse perception in Table 2). Therefore, we consider this a **minor limitation** that does not affect the main claims of our paper.
>
> ---
> **We sincerely welcome the opportunity to further discuss our work during the discussion phase and thank the reviewers for their valuable feedback.**

---

> > ### Comment · Reviewer_RksJ · 2025-08-03
> >
> > Thank you for the additional results and clarifications regarding training. I will raise my score accordingly.

---

> > > ### Author Response · Authors · 2025-08-04
> > >
> > > Thank you for your positive feedback and for recognizing our efforts in addressing your concerns. We sincerely appreciate your time and constructive comments, which have greatly helped improve our manuscript. We are glad to hear that the additional results and clarifications meet your expectations.

---

### Official Review · Reviewer_7V19 · 2025-07-03

**Clarity:** 3
**Significance:** 3
**Originality:** 3
**Rating:** 4
**Confidence:** 4

**Summary:**

The paper introduces NoisyGRPO, a reinforcement learning (RL) framework aimed at improving the chain-of-thought (CoT) reasoning in multimodal large language models (MLLMs). The method involves injecting controllable Gaussian noise into visual inputs to encourage exploration during RL, combined with a Bayesian advantage estimation that fuses this noise (as a prior) and observed rewards (as a likelihood) for more robust policy optimization. Through extensive experiments on established CoT quality, general ability, and hallucination benchmarks, the authors show that NoisyGRPO outperforms vanilla GRPO and other leading baselines, particularly enhancing the robustness and generalization of small-scale MLLMs like Qwen2.5-VL 3B.

**Questions:**

1. Methodological Differentiation: How does NoisyGRPO fundamentally distinguish itself from prior selective noise injection RL methods ([12]) and other recent exploration-boosting strategies in visual RL? What unique properties (empirical or theoretical) arise from using image-space noise + Bayesian estimation compared to similar approaches?
2. Reward Signal Robustness: Given that SBERT-based rewards may be imperfect for open-ended VQA, have you evaluated reward reliability or robustness directly? Are there scenarios where NoisyGRPO fails due to reward misspecification or bias?
3. Ablation on Scale and Efficiency: Can you provide concrete wall-clock time, GPU-hours, or memory usage for NoisyGRPO versus GRPO, especially on larger models, to support real-world deployment claims?
4. Broader Applicability: Is NoisyGRPO effective for other modalities or tasks (e.g., text-only, audio-visual tasks), or for LLMs not relying on visual input? What about transfer to reinforcement learning settings outside vision-language?
5. Error Analysis: For categories where NoisyGRPO underperforms (e.g., Coarse Perception, Coverage in AMBER), do you observe systematic failure modes? Can your Bayesian estimator be adapted or regularized to address these?

**Ethical Concerns:**

["NO or VERY MINOR ethics concerns only"]

**Final Justification:**

Thanks to the author for their effort in the rebuttal. I would advocate for acceptance.

**Limitations:**

The authors provide an explicit limitations section addressing key assumptions and failure modes. Additional discussion on scalability and reward signal bias would further strengthen this.

**Quality:**

3

**Strengths And Weaknesses:**

## Strengths
1. Methodological Innovation: The work presents a clearly articulated extension to GRPO: noise-injected exploration at the input level, paired with a principled Bayesian advantage estimator. The approach is well-motivated by the identified limitations of policy exploration and trajectory evaluation in MLLMs RL.
2. Solid Theoretical Foundation: The Bayesian estimator for advantage estimation is soundly motivated and derived (Section 3.3), where noise magnitude serves as a calibrated prior and reward as the observed likelihood, resulting in adaptive, statistically justified updates.
3. Comprehensive Experimental Evaluation: The experiments span multiple relevant benchmarks (e.g., MME-CoT for CoT quality, MMStar for general reasoning, and AMBER for hallucination) and consider both small- and large-scale MLLMs. Table 1 and Table 2 report detailed results, showing consistent gains especially for smaller models, validating the claims.
4. Results Analysis and Ablation: The paper provides insight through ablations (Figure 4) on the effect of both core method components and hyperparameters, as well as comparisons of noise distributions. Performance trends over training (Figure 3) provide further empirical support for the learning dynamics described.
5. Clear, Well-Organized Writing: Despite some dense sections, the paper is mostly well-structured, with equations, pipeline figures (see Figure 2), and comprehensive result tables and plots.
6. Practical Relevance: The findings address a genuine problem—improving generalization and robustness of RL-driven reasoning in MLLMs, which is valuable for both researchers and practitioners dealing with multimodal and open-ended settings.

---
## Weaknesses
1. Limited Novelty Relative to Recent Works: While the combination of noise injection and Bayesian estimation is methodologically appropriate, the paper does not demonstrate that the specific form of injected noise (Gaussian/diffusion in visual input) and the exact Bayesian estimator extend far beyond prior ideas of selective noise in RL [12] and group-based RL advantage estimation. The differentiation from similar recent approaches (especially those cited in [12], DAPO [51], and VLM-R1 [39]) could be clarified further.
2. Empirical Gains Are Sometimes Marginal: Though the main method outperforms baselines, in Table 2 the MMStar overall gains (“+2.8 and +2.6”) over the base model are present but not transformative. In some subcategories (e.g., Coarse Perception), NoisyGRPO underperforms GRPO, calling into question broad generality.
3. Complexity of the Bayesian Estimator: The approach introduces several new hyperparameters (e.g., $\alpha$, $\gamma$ in the Bayesian update; normalization steps) whose role becomes somewhat intricate (see Figure 4), requiring careful tuning. While the ablation explores sensitivity, practitioners may face difficulty generalizing the method without significant validation infrastructure.
4. Reliance on Embedding-Based Rewards: The SBERT-based reward may bias the optimization, and although the authors claim the estimator mitigates this, the evidence for such mitigation is limited to indirect observations. Section 3.1 concedes some rewards for open-ended tasks are imperfect, but analysis is not fully conclusive.
5. Scalability and Practical Impact: Although the authors claim low additional computational cost, there is no direct measurement or discussion (e.g., training time, inference cost increases due to noise sampling and additional rollouts). It would be beneficial to quantify this, especially for larger models where efficiency could be more critical.
6. Analysis Lacks Broader Perspective: The main results (Tables 1 and 2) focus on Qwen2.5-VL 3B and 7B variants, with only limited comparison to a wider array of models or tasks. Wider external generalizability or applicability remains somewhat speculative.
7. Interpretability of Results: Some key claims, such as the link between the proposed method and reduced hallucination (Table 2, AMBER benchmark), would benefit from deeper qualitative error analysis beyond metric scores.

---

> ### Author Rebuttal · Authors · 2025-07-30
>
> **Thank you for acknowledging the methodological soundness, theoretical rigor, and practical value of our work; we will address your concerns in detail below.**
>
> ---
> **Q1 & W1** *Methodological Differentiation.*
>
> ---
> Thank you for raising this important question.
> **Our method is conceptually distinct from prior RL works.**
> Below, we provide clarification on how our approach differs from prior methods such as SNI\[12], DAPO \[51], and VLM-R1 \[39]. We will add these discussions to the related works in the revised version:
>
> **Differentiation over Selective Noise Injection[12]**
>
> While both SNI [12] and our method leverage noise injection in policy exploration to enhance RL generalization, our approach fundamentally differs from SNI in two key aspects, as outlined below.
> - **Off-policy handling for noise-injected policy exploration.** SNI [12] adopts a manually controlled weighted sum of gradients from noisy and clean rollouts (Eq. 7 in [12]), using a *fixed mixing ratio set by a hyperparameter*.
> In contrast, our method models the controllable noise level as a prior within a Bayesian advantage estimation framework, enabling *trajectory-wise dynamic calibration* of advantages and more principled off-policy correction.
> - **Noise Injection Strategy.** SNI [12] applies noise indirectly through structural mechanisms like Dropout and Variational Information Bottleneck (VIB), which perturb intermediate feature representations. In contrast, our method injects noise directly into the input images during rollout generation—a vision-language-specific strategy that jointly enhances exploration and robustness to multimodal hallucination, which is not explored by prior methods.
>
> **Differentiation over DAPO and VLM-R1**
>
> While DAPO and VLM-R1 also build on the GRPO backbone, our method optimizes GRPO from a principled perspective. We provide justification as below:
> - **DAPO** proposes several improvements to enhance the stability and efficiency of training with GRPO, including Clip-Higher, Dynamic Sampling, Token-level Policy Gradient Loss, and Overlong Reward Shaping.
> - **VLM-R1** focuses on applying GRPO to REF and OVDet tasks, designing detection-specific reward functions tailored for these domains.
>
> In contrast, the main focus of our method lies in addressing the challenges of *insufficient exploration* and *the absence of CoT-process rewards* in multimodal RL. Beyond targeting different tasks, our approach introduces a *principled framework* that tackles these issues through controllable noise injection and a Bayesian modeling of advantage estimation, enabling more effective trajectory-level advantage assignment and policy exploration in multimodal settings.
>
> ---
> **Q2 & W4** *Reward Signal Robustness*
>
> ---
> Thanks for the insightful question! We first clarify our claim regarding the SBERT reward bias, then we address the concerns of reward robustness.
>
> **Clarification** We would like to clarify that our intention is **not to suggest** that the Bayesian estimator eliminates the bias inherent in the SBERT-based reward model. Rather, we aim to claim that **within our experimental setting**, which involves multimodal RL across verifiable and unverifiable data, **the Bayesian advantage estimation alleviates the optimization bias** introduced by noisy or misaligned reward signals, as emperically demonstrated in paper Section 4.6.
>
> **Reward Signal Robustness** The presence of bias in rewards for unverifiable data is **inevitable** in open-ended RL tasks. For instance, widely adopted solutions such as LLM-as-Judge [1] produce inconsistent or unreliable signals, as highlighted in [2].
>
> We choose SBERT as our reward model due to its **strong semantic understanding** and its **computational efficiency**. We also **empirically validate** its effectiveness, showing consistent gains with NoisyGRPO. We acknowledge that **SBERT’s bias may lead to failure in some cases**, but emphasize that SBERT is a **pluggable component** and can be replaced with stronger or task-specific reward models as they become available.
>
> Furthermore, we acknowledge that **reward for unverifiable data** is an important future direction. We are currently investigating principled ways to diagnose and improve insufficient exploration and missing process CoT reward issues in multimodal RL settings.
>
> Reference:
>
> [1] Rlaif vs. rlhf: Scaling reinforcement learning from human feedback with ai feedback.
>
> [2] Scaling Laws for Reward Model Overoptimization.
>
> ---
> **Q3 & W5:** *Ablation on Scale and Efficiency*
>
> ---
>
> Good suggestion! First, we clarify that NoisyGRPO and GRPO **sample the same number of rollouts**. The wall-clock time, GPU-hours, and peak memory usage for both methods are summarized in the table below to support real-world deployment considerations. The numbers are tested on one node with 8 A100 GPUs.
>
> | Model              | Method     | Wall-Clock Time | GPU-Hours      | Memory Usage     |
> |-|-|-|-|-|
> | Qwen2.5-VL-3B      | GRPO       | 6h12min          | 49.6 A100 hours | Peak GPU 57GB    |
> |                   | NoisyGRPO  | 6h40min          | 53.3 A100 hours | Peak GPU 53GB    |
> | Qwen2.5-VL-7B      | GRPO       | 7h58min          | 63.7 A100 hours | Peak GPU 75GB    |
> |                   | NoisyGRPO  | 7h52min          | 62.9 A100 hours | Peak GPU 77GB    |
>
> This suggests that NoisyGRPO maintains comparable efficiency with GRPO, despite its additional sampling step, making it suitable for real-world deployment.
>
> ---
> **Q4 & W6** *Broader Applicability*
>
> ---
>
> Thank you for the insightful suggestion. Our current work is tailored to **vision-language reinforcement learning**, where challenges like limited exploration and reward sparsity are prominent. While we have not yet applied NoisyGRPO to text-only or audio-visual tasks, the core idea of **injecting controllable perturbation to policy exploration and leveraging the perturbation level as prior for advantage estimation** is **modality-agnostic and potentially transferable**.  We view this as promising future work and hope to validate the method’s robustness and generality in broader settings.
>
> ---
> **Q5** *Error Analysis*
>
> ---
> We believe the observed limitations are due to benchmark-specific characteristics and constitute a minor issue in the broader context of our method’s performance.
>
> **Coverage in AMBER**
>
> We argue that **AMBER Coverage has inherent limitations** and does not provide evidence of a systematic failure in NoisyGRPO. In AMBER's generative setting, each case has on average **5.5 ground truth captions**, and our method shows at most a **2.7% decrease in Coverage** compared to the strongest baseline—equivalent to merely **0.14 missing words per case**, which we consider a negligible trade-off given the substantial gains in hallucination reduction.
>
> Moreover, we observe that **many hallucination-mitigating methods exhibit similar behavior** on AMBER, where improving factual accuracy can naturally lead to slightly lower coverage scores. This suggests that the metric may not fully capture the quality of grounded yet concise descriptions. To illustrate this, we propose a **CHR** to measure how much the hallucination rate decreases relative to the drop in coverage. A higher value indicates more effective hallucination reduction with less sacrifice in coverage.
>
> $\text{CHR} = \frac{|\Delta \text{Hallucination Rate}|}{|\Delta \text{Coverage}|}$
>
> This trend is not unique to our method — as the table shows, several recent strong baselines also experience similar trade-offs between hallucination reduction and coverage."
>
> | Method         | Cs ↓ | Cov. ↑ | Hal. ↓ | Cog. ↓ | CHR↑  |
> |-|-|-|-|-|-|
> | LLaVA-1.5 7B   | 7.8   | 51.0   | 36.4   | 4.2    | -    |
> | HA-DPO[1]        | 7.2   | 33.6   | 19.7   | 2.6    | 0.96 |
> | CLIP-DPO[2]       | 3.7   | 47.8   | 16.6   | 1.3    | 6.19 |
> | Qwen2.5-vl-3B  | 7.6   | 69.9   | 55.7   | 5.8    | -    |
> | NoisyGRPO 3B   | 6.6   | 67.7   | 44.3   | 3.4    | 5.18 |
> | Qwen2.5-vl-7B  | 4.8   | 63.6   | 27.5   | 1.6    | -    |
> | NoisyGRPO 7B   | 4.2   | 63.2   | 23.9   | 1.2    | **9.00** |
>
> From the table, we can see that other methods on AMBER also exhibit a decrease in hallucination accompanied by a drop in coverage, while NoisyGRPO 7B achieves the best CHR score. This suggests that **NoisGRPO reduces hallucination more effectively than others**, while incurring a relatively smaller drop in coverage, as reflected by the highest CHR score among all methods.
>
> Reference:
>
> [1] Beyond Hallucinations: Enhancing LVLMs through Hallucination-Aware Direct Preference Optimization, arXiv 2023, cited 148 times.
>
> [2] CLIP-DPO: Vision-Language Models as a Source of Preference for Fixing Hallucinations in LVLMs, ECCV 2024, cited 26 times.
>
> **Coarse Perception**
>
> We argue that this is **a minor issue** of NoisyGRPO, as our method achieves performance comparable to the baseline (within ±1% accuracy). This suggests that our approach remains effective even in challenging scenarios.
>
> The observed limitation stems from the fact that **our assumption — the monotonic relation between noise and trajectory quality — does not always hold** in Coarse Perception tasks. These tasks rely on high-level abstraction where fine-grained visual details are less critical, so noise injection may not sufficiently perturb the model's output to expose quality differences.
>
> However, **this does not undermine our method**, as the Bayesian advantage estimator is designed to adaptively downweight the prior when its confidence is low. Specifically, when noise fails to differentiate quality, the semantic reward's variance decreases (Eq. 8), leading to higher uncertainty in the prior and a reduced weight in the final estimation (Eq. 9). This adaptive mechanism ensures stable performance even when noise injection is less effective.
>
> ---
> **We sincerely welcome the opportunity to further discuss our work during the discussion phase and thank the reviewers for their valuable feedback.**

---

### Official Review · Reviewer_nP6a · 2025-07-07

**Clarity:** 2
**Significance:** 3
**Originality:** 3
**Rating:** 4
**Confidence:** 3

**Summary:**

The paper proposes NoisyGRPO, an RL training framework based on the classic GRPO method. Specifically, the method introduces (1) Noise-Injected Exploration Policy, which adds Gaussian noise to the images to obtain diverse rollouts, and (2) Bayesian Advantage Estimation, which takes the injected noise level as a prior, and the observed trajectory reward as the likelihood, to serve as advantage estimation.

**Questions:**

N/A

**Ethical Concerns:**

["NO or VERY MINOR ethics concerns only"]

**Final Justification:**

My major concerns have been solved. I think this has made the paper acceptable.

**Limitations:**

yes

**Quality:**

2

**Strengths And Weaknesses:**

## Strengths

1. The overall motivation is reasonable: when training RL, the rollouts tend to converge as the RL training proceeds. By injecting noise into images explicitly, the rollouts' diversity is ensured. The novelty is sound.
2. The reported model results seem to perform well across different benchmarks and different model sizes.
3. The paper is quite easy to read.

## Weaknesses

1. For the Off-Policy Strategy & Lack of Justification (L.161-162), the authors apply noise injection *only* during rollout collection but use clean inputs for policy updates. This creates an off-policy strategy where inference and training conditions differ. The paper lacks a clear justification for why this approach works effectively. More ablation studies and analysis are needed to validate this design choice.
2. The distinction between the two Bayesian advantage estimation strategies is unclear. One injects noise directly during advantage estimation, while the other injects noise into the image inputs during rollout collection. The paper fails to claim the fundamental difference or relative contribution of these two noise injection points.
3. The comparison with the SOTA methods ([1-3]) is insufficient. Performance should be rigorously evaluated against these methods using mainstream multimodal mathematical reasoning benchmarks like MathVista [4], MathVision [5], MathVerse [6], and MMMU Pro [7]. (MMStar is noted as containing only a small subset and is therefore not comprehensive enough for this purpose).
4. Inadequate Reward Function Analysis: The authors state they use SBERT to compute text similarity for open-ended answers instead of rule-based rewards (lines 143-146). However, they provide no ablation study demonstrating SBERT's superiority over traditional rule-based methods. Furthermore, no ablation analysis is presented for the threshold parameter `τ`.

[1] MM-Eureka: Exploring the Frontiers of Multimodal Reasoning with Rule-based Reinforcement Learning
[2] SoTA with Less: MCTS-Guided Sample Selection for Data-Efficient Visual Reasoning Self-Improvement
[3] NoisyRollout: Reinforcing Visual Reasoning with Data Augmentation
[4] MathVista: Evaluating Mathematical Reasoning of Foundation Models in Visual Contexts
[5] Measuring Multimodal Mathematical Reasoning with MATH-Vision Dataset
[6] MathVerse: Does Your Multi-modal LLM Truly See the Diagrams in Visual Math Problems?
[7] MMMU-Pro: A More Robust Multi-discipline Multimodal Understanding Benchmark

---

> ### Author Rebuttal · Authors · 2025-07-30
>
> **We appreciate the reviewer for the positive feedback and for recognizing the motivation, novelty, and clarity.
> We appreciate your comments and will address your concerns point-by-point below.**
>
> ---
> **W1:** *The Off-Policy Strategy & Lack of Justification*
>
> ---
> We sincerely thank the reviewer for this insightful and important question, which touches upon the core design motivation of our method. We address this concern through a three-step response:
> - We clarify how the policy update is conducted in our framework.
> - We justify our design by showing that it has already accounted for the mismatch between noisy rollouts and clean policy updates, in a way that aligns with the principle of importance sampling.
> - We present ablation studies comparing policy updates with and without noisy inputs, which confirm that using clean inputs for policy updates leads to more stable and effective learning.
>
> **Clarification of “using clean inputs for policy updates”**
> We provide the objective function of NoisyGRPO in Eq. (1), the policy gradient is computed based on the following objective:
>
> $$
> \nabla\_{\theta} J\_{\text{NoisyGRPO}}(\theta) =
> \mathbb{E}\left[
> \nabla\_{\theta} \log \pi\_{\theta}(o\_i^n \mid q, I) \cdot w\_i \cdot \tilde{A}\_i^k
> \right], \quad
> \text{where } w\_i = \frac{\pi\_{\theta\_{\text{old}}}(o\_i^n \mid q, I)}{\pi\_{\theta}(o\_i^n \mid q, I)}
> $$
>
> For clarity, we omit auxiliary components such as the KL penalty and gradient clipping in the above expression, which are standard in practice but do not affect the core analysis here.
>
> Here, *“using clean inputs for policy updates”* specifically refers to evaluating both the **policy gradient term** $\nabla\_{\theta} \log \pi\_{\theta}(o\_i^n \mid q, I)$ and the **importance weight** $w\_i $ under the clean image $I$.
>
> **Design Justification**
> Our Bayesian Advantage Estimation accounts for the mismatch between perturbed rollouts and clean policy updates, **serving a similar role to importance sampling** in off-policy learning.
>
> Specifically, our advantage $\tilde{A}^k_i$ is computed by normalizing a weighted sum of two reward estimates as shown in Eq. (9). Here, $\hat{r^n_i}$ is the group-normalized form of $r_i^n = 1-n_i$,  where $n_i$ denotes the noise level of the $i^{th}$ trajectory.
> And $\hat{r^n_i}$ could **approximate the importance sampling under our framework**.
>
> To aid understanding, we consider an extreme case for illustration, where we let $\tilde{A}^k_i = r_i^n$. In this case, the policy gradient update becomes:
> $$
> \nabla\_{\theta} J\_{\text{NoisyGRPO}}(\theta) =
> \mathbb{E}\left[
> \nabla\_{\theta} \log \pi\_{\theta}(o\_i^n \mid q, I) \cdot w\_i \cdot r\_i^n
> \right]
> $$
>
> We can reinterpret $r\_i^n$ as an importance weight:
>
> $r\_i^n  = \frac {1 - n} {1} = \frac {q(x)} {p(x)}$
>
> where $q(x)$ denotes the (clean) target distribution, and $p(x)$ denotes the noise-injected behavior distribution. This shows that **$r\_i^n$  downweights noisy trajectories proportionally to their estimated noise level $n^{i}$**, effectively approximating an importance sampling correction between $q(x)$ and $p(x)$.
>
>
> **Ablation Experiment**
> Additionally, we perform ablation experiments where the policy is updated using noise-injected images. As shown in the table below, this leads to a performance drop. This is because the evaluation scenario assumes clean input images, requiring the target policy to generate responses based on clean visual inputs. Therefore, **updating the policy with clean images during training ensures alignment with the evaluation setting**. These results validate our design choice.
>
>
> | Method                      | MMStar           |       |       |       |       |       |       | AMBER          |       |       |       |       | MMERealworld |
> |-|-|-|-|-|-|-|-|-|-|-|-|-|-|
> |                             | CP.  | FP.  | IR.  | LR.  | ST.  | MA.  | Avg.  | Cs ↓ | Cov. ↑ | Hal. ↓ | Cog. ↓ | F1 ↑  | Acc          |
> | Update policy with noise image | 67.0  | 49.3  | 57.6  | 55.2  | 35.1  | 59.3  | 53.9  | 7.2   | 66.3  | 45.1  | 4.2   | 88.7  | 36.6         |
> | Update policy with clean image  | 69.5  | 53.2  | 66.6  | 60.7  | 37.1  | 62.3  | 58.2  | 6.6   | 67.7  | 44.3  | 3.4   | 90.3  | 44.0         |
>
> ---
> **W2:** *The distinction between the two Bayesian advantage estimation strategies is unclear.*
>
> ---
> Apologies for the confusion. We would like to clarify that our method **does not** involve two separate Bayesian advantage estimation strategies. We only inject noise into the image inputs during the rollout collection phase. The Bayesian advantage estimation module does **not** involve any explicit noise injection.
>
> We recognize two potential sources of confusion that may lead to misunderstanding our approach:
>
> - **Noise prior** in line 192: The noisy prior is treated as a prior estimation of the trajectory quality decided by the noise level during rollout generation.
> - **Gaussian noise** in line 186: In our Bayesian estimator, we model the semantic reward $\hat{r}_s$ and noisy reward $\hat{r}_n$ as being corrupted by Gaussian noise, which enables us to obtain a posterior estimation $\hat{r}_i$ through Bayesian update.
>
> We hope this could resolve the confusion. We will also revise the manuscript accordingly in the revised version.
>
> ---
> **W3:** *Lack of comparison with the SOTA methods in multimodal mathematical reasoning benchmarks.*
>
> ---
>
> We thank the reviewer for raising this important point. Our work addresses a multimodal RL setting that is **orthogonal to** multimodal math reasoning, so extensive comparisons with multimodal math reasoning benchmarks are **beyond the scope of our work**.
>
> Specifically, prior works [1–3] on multimodal math reasoning assume **verifiable and unbiased reward signals** (e.g., exact answer matching), whereas our work addresses a **more realistic scenario** where reinforcement learning operates under potentially noisy or biased rewards—a setting we refer to as **RL beyond verifiable rewards**. This setting is crucial for real-world tasks such as visual chat and image captioning, where ground-truth signals are often ambiguous or unavailable. **Our method is specifically designed for this setting**, incorporating **Bayesian advantage estimation** to effectively mitigate the impact of reward noise during policy optimization.
>
> Furthermore, our study aims to **systematically investigate** whether **NoisyGRPO** can address **two under-explored but crucial challenges** in preference optimization for multimodal reasoning: (1) insufficient exploration during policy learning and (2) lack of supervision over intermediate reasoning steps. These aspects are **rigorously evaluated through controlled experiments** (see Table 1 and Table 2), where we measure both alignment with human preferences and improvement in reasoning consistency.
>
> While we acknowledge that a full comparison on multimodal math benchmarks would further validate our findings, we leave comprehensive benchmarking as future work and appreciate the reviewer’s suggestion in this direction.
>
> ---
>
> **W4:** *Inadequate Reward Function Analysis.*
>
> ---
> Thank you for highlighting the importance of reward design analysis.
>
> **Comparison with rule-based reward**
>
> To validate our choice of using the text embedding model SBERT as the reward model, we compare it with a popular rule-based reward approach. Specifically, we adopt the average F1 score of ROUGE‑1, ROUGE‑2, and ROUGE‑L as the reward. The results are shown in the table below.
>
> | Method                     | MMStar           |       |       |       |       |       |       | AMBER          |       |       |       |       | MMERealworld |
> |-|-|-|-|-|-|-|-|-|-|-|-|-|-|
> |                           | CP.  | FP.  | IR.  | LR.  | ST.  | MA.  | Avg.  | Cs ↓ | Cov. ↑ | Hal. ↓ | Cog. ↓ | F1 ↑  | Acc          |
> | Rouge reward  | 64.9 | 49.5 | 64.8 | 51.3 | 38.0 | 52.6 | 53.5  | 8.0   | 66.4   | 46.5   | 3.4   | 89.3  | 43.6         |
> | SBERT reward | 69.5 | 53.2 | 66.6 | 60.7 | 37.1 | 62.3 | 58.2  | 6.6   | 67.7   | 44.3   | 3.4   | 90.3  | 44.0         |
>
> As shown in the table, **SBERT consistently outperforms the rule-based reward** in our experimental setting.
> This improvement is likely due to SBERT’s ability to **capture semantic similarity beyond exact token overlap**, which is especially important for evaluating open-ended answers where lexical variance is common.
>
> **Hyperparameter $\tau$ Analysis**
>
> We first clarify that we set $\tau$ based on a reasonable default, and we humbly argue that introducing new hyperparameters is inevitable when proposing a new framework. Furthermore, we conduct an ablation study to show the sensitivity of $\tau$, and the results are presented in the table below.
>
> | Method | MMStar           |       |       |       |       |       |       | AMBER          |       |       |       |       | MMERealworld |
> |-|-|-|-|-|-|-|-|-|-|-|-|-|-|
> |        | CP.  | FP.  | IR.  | LR.  | ST.  | MA.  | Avg.  | Cs ↓ | Cov. ↑ | Hal. ↓ | Cog. ↓ | F1 ↑  | Acc          |
> | vanilla GRPO | 68.0  | 43.3 |  60.4 |  57.0  | 40.0  | 58.0  | 54.5  | 6.7  | 68.5  | 44.6  | 4.2 |  89.2 |  40.8 |
> | $\tau$=0.5  | 66.1 | 50.9 | 62.5 | 56.2 | 37.5 | 65.9 | 56.5  | 6.7   | 64.3   | 37.5   | 2.5   | 89.5  | 43.0         |
> | **$\tau$=0.6**  | 69.5 | 53.2 | 66.6 | 60.7 | 37.1 | 62.3 | 58.2  | 6.6   | 67.7   | 44.3   | 3.4   | 90.3  | 44.0         |
> | $\tau$=0.7  | 69.1 | 52.3 | 63.4 | 58.2 | 39.3 | 56.2 | 56.4  | 7.4   | 65.8   | 42.4   | 3.0   | 90.0  | 44.7         |
>
> The results demonstrate that the method’s performance remains stable over a reasonable range of $\tau$ values and all of the choices of $\tau$ outperform the baseline. These indicate robustness to the hyperparameter choice.
>
> ---
> **We sincerely welcome the opportunity to further discuss our work during the discussion phase and thank the reviewers for their valuable feedback.**

---

> > ### Comment · Reviewer_nP6a · 2025-08-05
> >
> > I would like to thank the authors for their efforts. The major concerns about the difference between inference and training have been solved. I strongly recommend that the authors update their final version based on these comments.
> > I would raise my scores.

---

> > > ### Author Response · Authors · 2025-08-05
> > >
> > > We sincerely appreciate your time and valuable feedback, which have significantly strengthened our work. We are grateful that our rebuttal addressed your major concerns. As recommended, we will carefully incorporate the comments into the final version of the manuscript.

---

### Note · Authors · 2025-08-15

Dear Reviewers and Area Chair,

We sincerely thank you for the valuable feedback and constructive discussion throughout the review process. We are encouraged by the supportive comments highlighting the strengths of our work:
- **Clear motivation & novel, principled framework** (Reviewers nP6a, 7V19, RksJ, TGQ6)
- **Sound theoretical foundation** (Reviewer 7V19)
- **Strong empirical performance with comprehensive evaluation & ablations** (Reviewers nP6a, 7V19, RksJ, TGQ6)
- **Clear presentation** (Reviewers nP6a, 7V19, RksJ, TGQ6)

Building on this positive feedback, we carefully addressed the reviewers’ concerns in our rebuttal:

- **Justification of Off-policy strategy.** We clarified the rationale behind the design of Bayesian Advantage Estimation and provided supporting results on the baseline mentioned by the reviewer, thereby justifying the effectiveness of our method’s design.
- **Analyzed the model's robustness to hyperparameters.** We conducted additional ablation studies demonstrating that our method remains robust to variations in both SBERT reward parameters and training hyperparameters, further validating its potential for real-world scenarios.
- **Ablation on Scale and Efficiency.** We reported concrete wall-clock time, GPU-hours, and memory usage of NoisyGRPO versus vanilla GRPO, showing that our method achieves better performance without incurring additional computational cost.

We are pleased that our rebuttal has fully and effectively addressed the concerns of Reviewers TGQ6, nP6a, and RksJ, and we are particularly encouraged by the willingness of Reviewers nP6a and RksJ to raise their scores. In the revised version, we will further refine the paper to enhance clarity and presentation.

- We will further improve the writing of the methods section by incorporating a clearer explanation from the perspective of the off-policy strategy.
- We will integrate all rebuttal experiments and analyses, including hyperparameter sensitivity analysis, ablations on scale and efficiency, and evaluations of generalization to text-only abilities.

---

### Decision · Program_Chairs · 2025-09-17

**Decision:**

Accept (poster)

**Comment:**

The paper introduces NoisyGRPO, an extension to GRPO which introduces a noise to image inputs to improve generalization, robustness, and exploration in CoT reasoning in MLLMs. A bayesian advantage estimator down weights noisy trajectories for policy optimization. Overall, the work is timely given attention on research on multimodal reasoning, and the approach is intuitive and sound. There were various questions regarding the noise prior and advantage estimation, as well as a variety of other concerns wrt comparisons to prior works, additional ablations, etc. However, the authors seem to have provided a very satisfactorily rebuttal, and the reviewers (and I) were satisfied with the answers.

I therefore recommend the paper is accepted as a poster.